# INTRA-REQUEST BRANCH ORCHESTRATION FOR EFFICIENT LLM REASONING

## ABSTRACT

LLMs increasingly rely on inference-time reasoning algorithms such as chain-of-thought and multi-branch reasoning to improve accuracy on complex tasks. These methods, however, significantly increase token usage (cost) and per-request latency. Prior work has primarily focused on reducing token usage, often at the expense of accuracy, while overlooking other latency factors.

We present DUCHESS, an LLM serving system that reduces computational cost and latency without sacrificing accuracy through intra-request branch orchestration guided by predictions. Within each request, DUCHESS predicts branch correctness with a lightweight linear probing model over LLM layer activations. The orchestration policy uses these predictions to decide whether to terminate a branch early, duplicate an existing branch, or continue exploring a branch. When handling multiple requests, DUCHESS can further reduce latency by prioritizing easier reasoning tasks, when request complexity can be estimated from the prompt.

Experiments on three reasoning benchmarks show that DUCHESS consistently improves the token–accuracy Pareto frontier, reducing token usage by 42–63% at matched accuracy compared to self-consistency. For request serving with vLLM, DUCHESS reduces mean, median, and tail latencies by 57-81%, 58-85%, and 52-84% with First-Come-First-Served (FCFS) scheduling across three datasets, compared to self-consistency. At higher request rates, scheduling jobs by increasing predicted difficulty reduces mean latency by 25.1-29.7% further over FCFS.

## 1 INTRODUCTION

Large Language Models (LLMs) are widely applied across domains, including math and science problem-solving (Lewkowycz et al., 2022), coding and program analysis (Jiang et al., 2025), logical deduction (Creswell et al., 2023), and decision-making (Yao et al., 2023). Solving these tasks often requires *reasoning algorithms*, which extend beyond one-shot prediction. Reasoning algorithms (Wang et al., 2023; Wei et al., 2022; Yao et al., 2023; Wu et al., 2024; Snell et al., 2025) vary in implementation but share the goal of improving LLM reasoning at inference time. They enable models to reflect on their outputs, explore alternative reasoning paths, and produce more accurate answers. Among these methods, multi-branch reasoning (Wang et al., 2023) has become a widely used approach. The model generates multiple candidate reasoning paths (branches) for a request and aggregates their outcomes to produce the final answer. While individual branches may use chain-of-thought (Wei et al., 2022) reasoning, the defining feature of multi-branch reasoning is the exploration of multiple paths in parallel.

While reasoning algorithms improve accuracy, they also introduce substantial computational overhead. Long chains of thought and multiple branches increase token generation per request, and each additional token extends processing time. As LLMs are increasingly deployed in interactive and large-scale applications, these inefficiencies directly impact scalability and user experience. Building efficient LLM reasoning systems, therefore, requires balancing cost, latency, and accuracy, which gives rise to two main challenges.

The first challenge is the cost–accuracy tradeoff: pruning output tokens reduces inference cost but can harm accuracy. For example, terminating a branch's reasoning too early (Zhang et al., 2025; Afzal et al., 2025; Yang et al., 2025; Liu & Wang, 2025; NVIDIA, 2025) risks missing the correct answer, while stopping too late wastes tokens. In multi-branch reasoning, accuracy also depends on

how different branches interact, yet existing methods often treat branches independently and optimize only for cost (Hassid et al., 2025; Li et al., 2024; Fu et al., 2024). Beyond this cost–accuracy tradeoff, a second challenge arises in maintaining low latency under concurrent serving, where multiple reasoning requests compete for shared resources. Latency depends not only on the number of generated tokens but also on system-level factors such as queuing delays, parallel execution efficiency, and straggler branches that determine request completion time. Methods that reduce token counts, such as adaptively generating branches (Li et al., 2024), can be less parallelizable and therefore slow down throughput. Other approaches rely on complex orchestration—using an auxiliary model (Jin et al., 2023; Leviathan et al., 2023) or reinforcement learning (Aggarwal & Welleck, 2025; Dai et al., 2025; Hou et al., 2025; Lou et al., 2025) which improves token efficiency but introduces additional overhead that may increase end-to-end latency.

In this paper, we present DUCHESS, an LLM serving system that reduces cost and latency while maintaining accuracy. Given a reasoning request, DUCHESS orchestrates its branches to complete it as efficiently as possible. It applies an intra-request branch orchestration policy guided by predictions derived from LLM layer activations. These predictions estimate the correctness of each branch and inform three possible branch-level actions:

- Early termination: stop a branch once it is likely to already yield a correct answer, avoiding unnecessary token generation.

- Selective branch-out: duplicate a promising branch to increase coverage while reusing its past progress, rather than starting from scratch.

- Continuation: keep generating tokens for branches that are still uncertain, giving them additional steps to potentially reach a correct answer.

These branch-level decisions reduce token usage with minimal accuracy loss. Additionally, the system terminates the entire request once either a consensus among enough branches is reached or a sufficient number of answers are collected, preventing straggling branches from delaying completion.

At the request-pool level, DUCHESS can optionally apply a lightweight scheduling step before intra-request orchestration. If request difficulty can be estimated from the prompt, DUCHESS prioritizes easier requests first, following the intuition of Shortest Job First (SJF) and prior works on LLM serving systems (Shahout et al., 2025; 2024; Fu et al., 2024; Mitzenmacher & Shahout, 2025). Otherwise, it defaults to First-Come-First-Served (FCFS). This scheduling is not central to our approach but can offer additional latency benefits when complexity information is available.

We evaluate DUCHESS using three popular reasoning benchmarks datasets: GSM8K (Cobbe et al., 2021), MMLU (Hendrycks et al., 2021a), and MATH (Hendrycks et al., 2021b) and on three LLMs: DeepSeek-R1-Distill-Llama-8B (Guo et al., 2025), Intern-S1-Mini (Bai et al., 2025), and Gemma-3-4B-IT (Team et al., 2025). Our results demonstrate that DUCHESS efficiently reduces reasoning cost and latency without trading accuracy. On R1-Distill-Llama, DUCHESS dominates the cost-accuracy Pareto frontier in all of our experiments, reducing token usages by 42%-63% at matched accuracy compared to Self-consistency (Wang et al., 2023), a popular multi-branch reasoning algorithm. For request serving with vLLM (Kwon et al., 2023), DUCHESS reduces mean, P50, and P95 latencies by 57-81%, 58-85%, and 52-84% on three datasets using First Come First Served (FCFS). Furthermore, at higher arrival rates, applying complexity-aware scheduling reduces DUCHESS mean latency by up to 25.1-29.7% and 25.6-34.7% compared to FCFS on the three datasets, using predicted and actual difficulty labels, respectively.

## 2 BACKGROUND AND RELATED WORKS

**Single CoT token reduction.** Recent works have studied pruning a single CoT sequence to reduce token usage, and/or avoid accuracy drops caused by overthinking. (Zhang et al., 2025; Afzal et al., 2025) predicts future CoT correctness based on current LLM hidden states; (Yang et al., 2025) identifies transit points in reasoning processes such as "wait" for potential early exit opportunities. (Liu & Wang, 2025; Fu et al., 2024) relies on intermediate answer consistency for CoT convergence. Overall, these works do not consider cross-branch dependencies within a request, which can compromise the optimality of cost-accuracy tradeoffs.

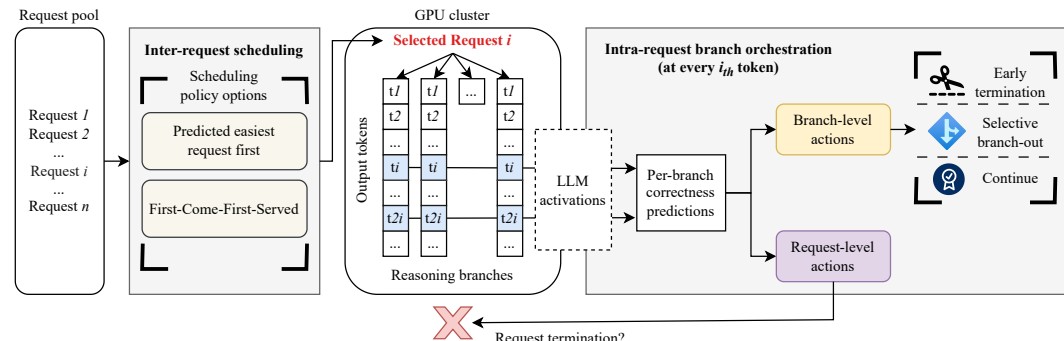

Figure 1: **DUCHESS overflow.** DUCHESS schedules requests by prioritizing easier ones when complexities can be estimated from the prompt, otherwise defaulting to First-Come-First-Served. To process a single request, at every $i_{th}$ token, DUCHESS predicts branch correctness from LLM activations and decides: at the branch level, whether to terminate, branch out, or continue. At a request level, the decision is made on whether to terminate the entire request once either a suitable consensus is reached or a sufficient number of answers are collected.

**Multi-branch token reduction.** In contrast to the above, some works jointly optimize all branches within the request, which usually delivers more optimal cost-accuracy tradeoffs due to reasoning awareness but sometimes overlook latency requirements. Incremental self-consistency methods (e.g., ESC (Li et al., 2024), RASC (Wan et al., 2025), Adaptive Consistency (Aggarwal et al., 2023)) generate branches in small batches until certain confidence conditions. These works can achieve good token-accuracy tradeoffs, but are unable to be sufficiently parallelized. Some use complex solutions (e.g., another transformer/LLM (Jin et al., 2023; Leviathan et al., 2023) or reinforcement learning methods (Aggarwal & Welleck, 2025; Dai et al., 2025; Hou et al., 2025; Lou et al., 2025)) to orchestrate request serving or reasoning progress at high additional overhead. (Hong et al., 2025) prunes reasoning branches based on similarities.

## 3 METHOD

DUCHESS reduces the computational cost and latency of long CoT multi-branch reasoning requests without sacrificing accuracy. At each step, DUCHESS selects a request from the pool and completes it as quickly as possible to reduce wait time. Within each request, DUCHESS applies an intra-request branch orchestration policy: after every $i^{th}$ token, it uses a lightweight linear probing model on transformer activations to estimate per-branch correctness probabilities. Based on these predictions, DUCHESS determines both branch-level and request-level actions. At the branch level, it assigns each branch one of three actions: 1) early termination, 2) selective branch-out, or 3) continuation. This policy avoids wasted computation on low-value branches, producing accurate answers with fewer tokens while preserving branch-level parallelism to reduce latency. At the request level, since multi-branch reasoning relies on majority voting, DUCHESS terminates an entire request once either a majority consensus is reached or a sufficient number of answers are collected, preventing stragglers from delaying completion.

When scheduling across multiple requests, DUCHESS can optionally apply a complexity-aware policy that prioritizes easier requests if their difficulty is known or can be estimated from the prompt. Following the intuition of Shortest Job First (SJF), this step helps reduce average latency across requests. Figure 1 illustrates DUCHESS's workflow.

### 3.1 PER-BRANCH CORRECTNESS PREDICTION.

In this section, we describe our method for estimating branch correctness from transformer activations. Specifically, we employ a linear probing model to predict per-branch correctness, i.e., the probability that a correct answer can be produced from the branch's current CoT steps.

**Problem definition.** Given a prompt $p$ and a partially decoded branch consisting of tokens $t_1, t_2, \ldots, t_k$, we aim to estimate whether the LLM can already produce a correct answer based

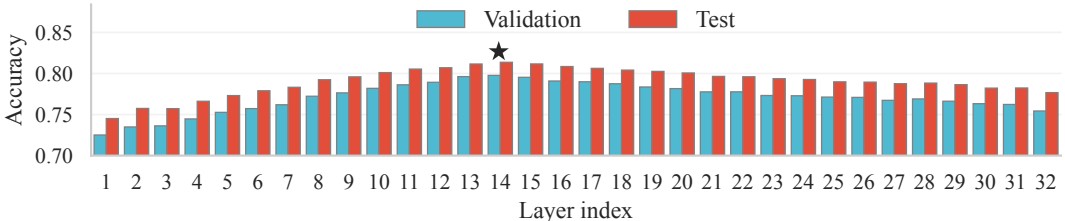

Figure 2: **LLM activations produce accurate correctness predictions.** The validation and test accuracies of the proposed MLP trained on each LLM transformer layer, using DeepSeek-R1-Distill-Llama-8B model and GSM8K dataset. Layer 14 yields highest validation and test accuracies.

on this partial sequence. If so, we terminate the branch and apply CoT probing (Fu et al., 2024), which queries the LLM for a direct answer using the existing tokens with the following probing prompt:

$$p + t_1 t_2 \ldots t_k + \texttt{``**Final Answer**\textbackslash n\textbackslash n\textbackslash[ \textbackslash boxed\{''}$$

The final segment instructs the LLM to output a direct answer conditioned on the tokens $t_1 \ldots t_k$.

In this work, we use CoT probing to extract answers from incomplete reasoning sequences. This technique requires only a small token budget (e.g., about 10 tokens) since the probed output is a short direct answer. (Fu et al., 2024) proposes applying CoT probing every 32–64 tokens and using certainty, whether probed answers remain stable across consecutive intervals, as a proxy for correctness. DUCHESS, in contrast, predicts correctness probabilities directly from the transformer activations of the last decoded token $t_k$. Since $t_k$ encodes the reasoning progress up to that point, its activations provide informative features for correctness prediction. These activations are produced during normal decoding, so our method adds no extra collection or processing overhead.

**Predictor design.** We design the predictor as a multi-layer perceptron (MLP). The input is the activation of token $t_k$ from one of the LLM's embedding layers. Although using more layers could improve accuracy, it would also increase computational overhead; therefore, we use a single layer for simplicity. The MLP outputs a single logit representing the probability that a CoT probing after token $t_k$ would yield a correct answer. We implement and train the predictor in PyTorch (Paszke et al., 2019). To account for dataset-specific characteristics, we vary other architectural choices (e.g., number of layers, hidden-layer widths) and training hyperparameters (e.g., learning rate, class weighting) per dataset. Full configurations appear in Appendix A.1.

**Data collection and predictor training.** We construct the training set from <prompt, groundtruth> pairs. For each prompt, we generate 10 LLM responses, each capped at 4,096 tokens. From every response, we apply CoT probing at intervals of 16 tokens to extract intermediate answers. Smaller intervals produce more training samples but also increase overhead; empirically, probing per 16 tokens provides sufficient training samples at a manageable cost. We assign a positive label to token $t_i$'s activation if three consecutive intermediate answers are correct starting at $t_i$, which filters out "lucky guesses" where correctness fluctuates. The MLP uses the activation of $t_i$ as input features and the correctness label as the target. We train the MLP with cross-entropy loss and the AdamW optimizer. We evaluate this approach on datasets such as GSM8K, MMLU and MATH, and train the predictor separately on each dataset's training split.

Figure 2 reports validation and test accuracy when training on activations from different LLM layers, using the GSM8K dataset and DeepSeek-R1-Distill-Llama-8B model. The middle layers yield the highest accuracy, peaking at layer 14, consistent with prior findings that middle layers capture the richest information in a transformer (Shahout et al., 2025; Skean et al., 2024). Due to time and resource constraints, we use layer 14 activations for all datasets tested with R1-Distill-Llama. Ideally, we would perform per-layer profiling for each dataset and LLM configuration, and we consider this a natural direction for future work.

### 3.2 INTRA-REQUEST BRANCH ORCHESTRATION

DUCHESS orchestrates a single request using predictions of branch correctness. After every $i$ tokens per branch, it applies the activation-based MLP to estimate the probability that each branch can yield a correct answer. Smaller intervals $i$ provide finer-grained monitoring but increase prediction overhead. In practice, we found $16 \leq i \leq 80$ to be a good tradeoff.

Based on these predictions, DUCHESS assigns branch-level actions and, when applicable, request-level actions.

**Branch-level actions**  At the branch level, DUCHESS supports three actions based on predicted correctness: (1) early termination, (2) selective branch-out, and (3) continuation.

**1. Early termination.** DUCHESS terminates a branch if its predicted correctness exceeds a threshold $\tau$ for $S$ consecutive rounds. We set $\tau$ to the 70-80th percentile of validation predictions and $S = 2$ to prevent premature termination. When a branch is terminated, DUCHESS applies CoT probing to extract a final answer from its current progress. If the branch ends naturally (EOS token), its final answer is retained without probing.

**2. Selective branch-out.** Terminated branches free GPU slots. Rather than start new branches from scratch, which risks stragglers, DUCHESS forks a new branch from an existing one, reusing its CoT progress and KV cache to reduce overhead. When there are $r$ active branches with correctness probabilities $p_1, \ldots, p_r$, DUCHESS samples according to a rescaled distribution:

$$\hat{p}_j = \frac{p_j^{1/\lambda}}{\sum_{k=1}^{r} p_k^{1/\lambda}}, \quad 1 \le j \le r,$$

where $\lambda$ is a temperature parameter. Smaller $\lambda$ favors high-confidence branches, while larger $\lambda$ increases the chance of duplicating lower-confidence ones. We tune $\lambda$ on validation data.

**3. Continuation.** Branches that are not terminated continue decoding for another $i$ tokens before being re-evaluated.

**Request-level action**  Since multi-branch reasoning relies on majority voting, DUCHESS terminates the entire request once enough answers are available. We propose two conditions: (1) at least $\alpha \cdot c$ identical answers are collected, or (2) at least $\beta \cdot c$ answers are collected, where $0 < \alpha < \beta < 1$ and $c$ is the number of parallel branches. The first condition captures fast consensus, while the second ensures sufficient coverage. Smaller $\alpha$ and $\beta$ values lead to more aggressive termination at the cost of potential accuracy loss.

**Parallelism.** Additionally, DUCHESS maintains up to $c$ active branches at all times. In practice, $c$ can be set based on maximum batch size allowed by hardware. This design ensures that GPU resources are fully utilized and prevents idle slots, thereby reducing per-request latency.

**Hyperparameter configuration and tuning.** We discuss the hyperparameter choices, selection criteria, and auto-tuning procedures in Appendix A.2.

### 3.3 INTER-REQUEST COMPLEXITY-AWARE SCHEDULING.

In pooled serving, prioritizing requests with shorter runtimes reduces mean latency. For LLM requests, runtimes must be predicted because autoregressive decoding is non-deterministic. Existing work (Shahout et al., 2025; Jin et al., 2023; Zheng et al., 2023) focuses on predicting output token lengths; however, actual runtimes also depend on algorithmic factors (e.g., early termination) and system factors (e.g., prefix caching), making them harder to estimate.

We observe that DUCHESS naturally completes easier requests faster. For example, in the MATH dataset (Hendrycks et al., 2021b), where requests are labeled with difficulty levels from 1 (easy) to 5 (hard), as difficulty increases, so does the average time DUCHESS spends per request: 13.8, 18.4, 25.8, 33.6, 47.4 for levels 1-5 (using 10 branches). Motivated by this, we introduce an optional complexity-aware scheduling policy: after completing one request, DUCHESS selects the easiest available request from the queue, provided difficulty labels are known. Otherwise, DUCHESS defaults to FCFS.

While the MATH dataset has labels, generally difficulty labels will need to be predicted. We therefore replaced the exact labels in the MATH dataset with predicted labels (using the original labels for training). Leveraging the informativeness of LLM activations, we developed a simple MLP model with three hidden layers for request complexity prediction using the MATH dataset. The input to the MLP is the activation of the first decoded token (i.e., after prefilling stage) from layer 14 of R1-Distill-Llama, and the output is the probability distribution across 5 complexity levels. For datasets that do not provide request complexities for training (e.g., GSM8K, MMLU), we use LLM-generated request complexities as groundtruth labels to train the predictor (detailed in A.3).

When using predicted complexities to schedule requests, DUCHESS first performs prefilling for all newly arrived requests, since the predictor relies on prefilling activations as input. Although the prefilling increases request queueing delays, its cost is insignificant relative to decoding, ensuring that DUCHESS still delivers end-to-end latency reductions.

We show in Appendix section A.3 the detailed predictor design, and the confusion matrix for predicted request complexities. In summary, we observe our predictor is able to differentiate easier and harder requests well, which has been shown to reduce queueing delays (Mitzenmacher, 2021; Mitzenmacher & Shahout, 2025).

## 4 EVALUATION

### 4.1 EXPERIMENTAL SETTINGS.

**Datasets and model.** We evaluate DUCHESS on three popular reasoning benchmarks: GSM8K (Cobbe et al., 2021), MMLU (Hendrycks et al., 2021a), MATH (Hendrycks et al., 2021b). We use the official train, validation, and test splits of the MATH dataset. For the other datasets, we use their official test splits, and randomly partition the train split into train and validation sets due to the lack of official validation splits, following approximate 70-10-20 ratios. We evaluate our approach using three LLMs, including two reasoning LLMs: DeepSeek-R1-Distill-Llama-8B model (Guo et al., 2025), Intern-S1-Mini (Bai et al., 2025), and one non-reasoning LLM: Gemma-3-4B-IT (Team et al., 2025), to demonstrate generalizability across broad model classes. We include results on DeepSeek-R1-Distill-Llama-8B in the main text, and results on the other two LLMs in Appendix for space considerations.

**Metrics.** Our primary metrics are: **(1) Cost-accuracy tradeoff.** We vary the maximum number of branches per request $5 \leq c \leq 10$, close to settings used by related works (Hassid et al., 2025). For each $c$, we record inference cost (i.e., the average number of output tokens generated per request) and the average accuracy across all requests. The max token size per branch is capped at 4,096. Our goal is to achieve better Pareto optimality, that is, using fewer tokens at matched accuracy, and achieving higher accuracy at matched tokens, compared to existing systems. **(2) Mean/P50/P95 latencies and Time-To-First-Tokens (TTFT).** We randomly generate request arrival schedules following Poisson distributions with an average rate of 2 queries per minute (QPM) for GSM8K and MMLU, and 1 for MATH. These rates are determined based on the upper quantiles of default self-consistency processing time per request at $c = 10$ (26.5, 30.5, and 64.0 seconds, respectively), thereby reflecting realistic loads in a single-server setting. We report the mean, P50, and P95 latencies (time between request arrival and completion) and TTFTs (time between request arrival and first decoded token): the former reflects end-to-end runtime, whereas the latter emphasizes queueing delays. We use vLLM v0.7.3 (Kwon et al., 2023) with chunked prefilling and prefix caching enabled,

**Baselines.** We compare DUCHESS to: **(1) Default self-consistency** (Wang et al., 2023) with no token-saving efforts. **(2) Dynasor** (Fu et al., 2024) for single branch early termination, which each branch is prompted to generate an intermediate answer at fixed token intervals, and early terminated if a sufficient number of consecutive consistent intermediate answers are observed. **(3) Short-m@k** (Hassid et al., 2025) for request-level early termination, which generates $k$ branches in parallel and terminates the whole request after the first $m$ out of $k$ finish, handling straggler effects at the cost of fewer final answers. For (2) and (3), we use the default settings from the original papers if possible, and only adjust them to achieve comparable accuracy levels in certain cases.

**Hardware and system settings.** We measure serving latencies on a testbed with one NVIDIA-A100 80GB GPU and 64 AMD EPYC 7313 16-Core Processors. We use vLLM v0.7.3 (Kwon et al., 2023) with `enable-prefix-caching` set to true; this ensures KV reuse in forked branches and in answer extraction from early-terminated branches. We use matched sampling parameters to (Fu et al., 2024), with 0.6, 0.95, and 1 for temperature, top p, and n, respectively.

### 4.2 EVALUATION RESULTS.

**DUCHESS dominates the cost-accuracy Pareto frontier across benchmarks** (Figure 3). DUCHESS consistently achieves higher accuracy at the same token budget and lower token usage at the same accuracy, strictly outperforming baselines in all of our experiments. For example, while matching default SC's max accuracy (within 0.1% tolerance), DUCHESS reduces token usage

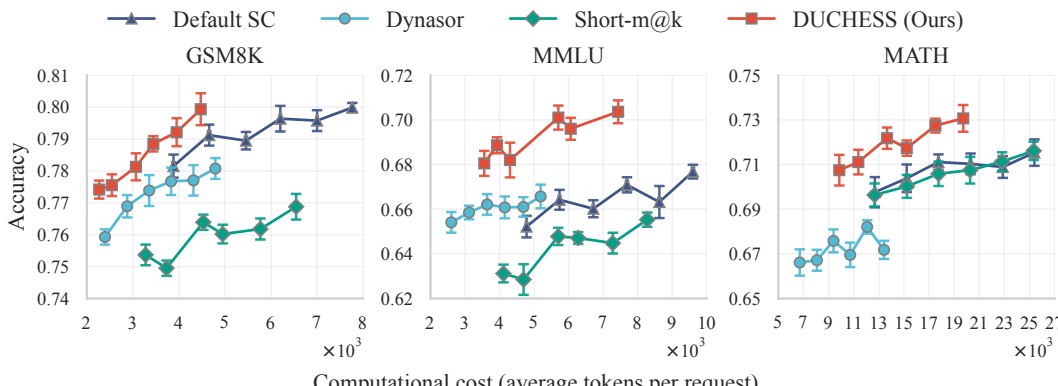

Figure 3: **DUCHESS reduces computational cost without compromising accuracy.** Cost-accuracy tradeoffs with branches per request varying between 5 and 10 inclusively. Markers within a curve correspond to increasing number of branches per request from left to right. The results are averaged from 10 trials, with error bars representing 95% confidence intervals.

| Method | GSM8K | | | MMLU | | | MATH | | |
|---|---|---|---|---|---|---|---|---|---|
| | **Mean** | **P50** | **P95** | **Mean** | **P50** | **P95** | **Mean** | **P50** | **P95** |
| **Latency (s)** | | | | | | | | | |
| Default SC | 54.1 | 40.8 | 156.1 | 82.8 | 62.7 | 215.3 | 137.7 | 117.3 | 319.5 |
| Short-m@k | 15.7 | 10.0 | 44.3 | 23.9 | 14.7 | 72.2 | 69.9 | 60.0 | 182.9 |
| DUCHESS | 10.5 | 7.7 | 24.7 | 17.0 | 9.2 | 66.7 | 58.8 | 48.8 | 154.4 |
| **TTFT (s)** | | | | | | | | | |
| Default SC | 34.2 | 16.1 | 134.5 | 59.7 | 39.9 | 191.0 | 88.2 | 63.7 | 260.3 |
| Short-m@k | 5.1 | 0.070 | 27.4 | 10.3 | 0.072 | 52.3 | 33.8 | 10.8 | 142.1 |
| DUCHESS | 2.8 | 0.068 | 14.4 | 6.7 | 0.067 | 43.7 | 27.6 | 2.21 | 118.6 |

Table 1: **DUCHESS reduces request latecies and TTFTs.** Mean/P50/P95 latencies and time-to-first-token (TTFT) in seconds, measured from one run using 10 branches per request. The accuracies of Default SC/Short-m@k/DUCHESS are 0.80/0.77/0.80 on GSM8K, 0.67/0.66/0.70 on MMLU, 0.71/0.70/0.73 on MATH. Requests follow a Poisson arrival process with mean rates of 2, 2, and 1 requests/min for GSM8K, MMLU, and MATH, respectively.

by 42%, 63%, and 46% respectively across three benchmarks, demonstrating significant inference cost savings. Notably, at $c = 10$, DUCHESS achieves 2.7% and 1.5% higher accuracy than Default SC on MMLU and MATH datasets, respectively. This is due to DUCHESS's early termination improving per-branch accuracy rates, which we discuss more in section 4.5.

Existing works that only apply early termination to reasoning branches reduce accuracy compared to Default SC, highlighting the advantage of branch orchestration. Dynasor consistently reduces tokens but drops max accuracy by 1% to 5% across all tested datasets and number of branches per request. This shows that intermediate answer consistency cannot always indicate correctness. Short-m@k reduces max accuracy across all $c$s by 2% and 3% on GSM8K and MMLU datasets. We observe that shorter branches in several requests are more likely to be incorrect due to underthinking, which the Short-m@k is biased towards. In addition, Short-m@k has negligible token usage reduction (0.2% at $c = 10$) on the MATH dataset because branch lengths are similar in most requests, leaving little room for pruning.

**DUCHESS reduces Mean/P50/P95 latencies and TTFTs in pooled request serving.** Table 1 reports latency and TTFT numbers with 10 branches per request. Across three datasets, DUCHESS reduces mean, P50, and P95 latencies by 57-81%, 58-85%, and 52-84%, respectively. DUCHESS achieves even larger reductions in TTFTs, with 69-92%, 96-99%, and 54-89% in mean, P50, and P95 cases, as a result of significantly lower queueing delays. We include Short-m@k measurements for reference, although its accuracies are often unmatched to Default SC (shown in Figure 3).

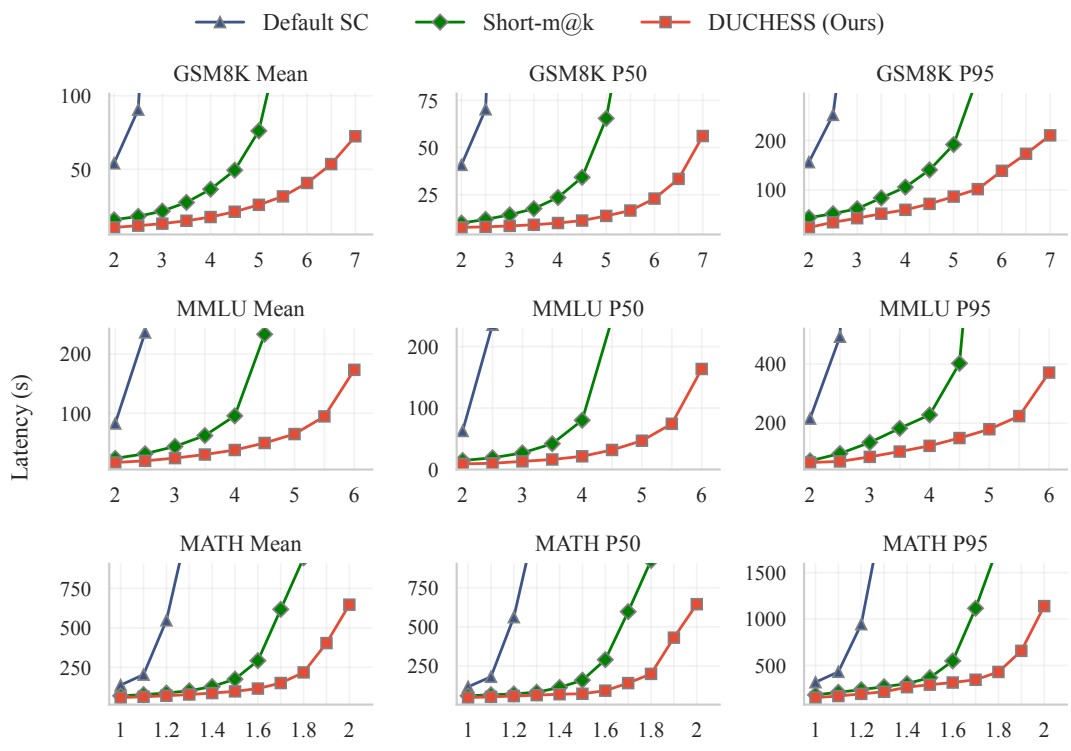

Figure 4: **DUCHESS is robust to bursts of requests.** Mean, P50, and P95 latencies as a function of request arrival rate (10 branches per request).

**DUCHESS is robust to bursts of requests.** Figure 4 shows the mean, P50, and P95 latencies as a function of request arrival rate, using up to $2\text{-}3.5\times$ higher QPM than rates in Table 1. Other settings remain unchanged. DUCHESS delivers the lowest latencies across reported request loads. With matched mean latency to Default SC at 1.5 QPM, DUCHESS can process $3\times$, $2.5\times$, and $1.6\times$ more requests across three datasets, demonstrating robustness to bursts of requests. We also observe that Short-m@k also substantially reduces latency compared to Default SC (although at smaller magnitudes than DUCHESS), highlighting the latency inflation caused by straggler branches.

**DUCHESS generalizes to additional reasoning and non-reasoning LLM classes and datasets.** We report DUCHESS's cost-accuracy tradeoffs and latency along with TTFT measurements on additional LLMs and datasets in Appendix A.4. Overall, DUCHESS reduces cost and latency while maintaining accuracy across diverse use cases, demonstrating generalizability and practicability for real-world deployment.

### 4.3 INTER-REQUEST COMPLEXITY-AWARE SCHEDULING.

Figure 5 shows the mean latencies as a function of request arrival rates, for DUCHESS using FCFS and complexity-aware scheduling with predicted and groundtruth complexities, using R1-Distill-Llama at 10 branches per request. The groundtruth request complexities are LLM-labeled for GSM8K/MMLU, and are provided for MATH. We increased the request arrival rate until the FCFS mean latency reached approximately $2e3$ seconds.

**Complexity-aware scheduling lowers mean latency relative to FCFS, with larger gains at higher request rates.** Across QPM values tested, complexity-aware scheduling reduces mean latency by 8.2-25.6%, 21.0-32.7%, and 10.4-34.7% using groundtruth request complexities, and by 5.3-25.1%, 19.3-26.2%, and 14.2-29.7% using groundtruth request complexities, on GSM8K, MMLU, and MATH datasets, compared to FCFS. As QPM increases, request queues grow, amplifying the benefits of complexity-aware scheduling.

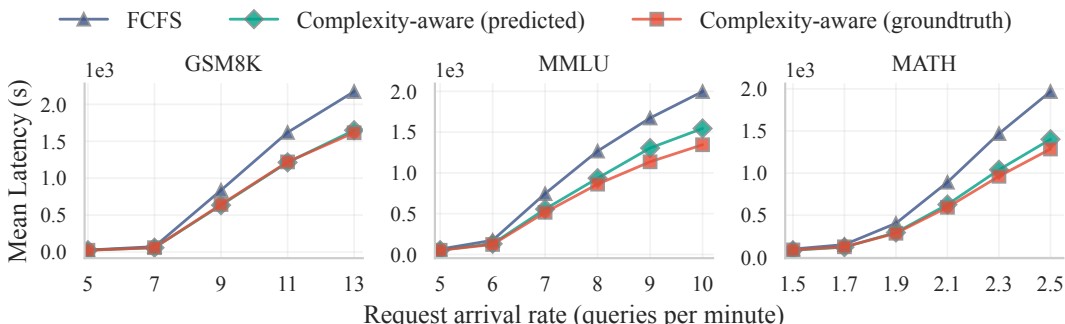

Figure 5: **Complexity-aware request scheduling reduces mean latency compared to FCFS.** Mean latency as a function of request arrival rate using FCFS, complexity-aware using predicted and groundtruth request complexities (10 branches per request). Groundtruth request complexities are LLM-labeled for GSM8K/MMLU, and are provided for MATH.

| | Cost–Acc | | Latency (s) | | | TTFT (s) | | |
|---|---|---|---|---|---|---|---|---|
| **Method** | **Cost** | **Acc.** | **Mean** | **P50** | **P95** | **Mean** | **P50** | **P95** |
| Default SC | 7763 | 0.80 | 54.1 | 40.8 | 156.1 | 34.2 | 16.1 | 134.5 |
| ESC | **2861** | **0.83** | 147.1 | 85.1 | 454.9 | 123.5 | 60.0 | 432.7 |
| DUCHESS (Ours) | 4474 | 0.80 | **10.5** | **7.7** | **24.7** | **2.8** | **0.1** | **14.4** |

Table 2: **Incremental Self-Consistency increases latency and TTFTs on GSM8K/R1-Distill-Llama.** The cost-accuracy tradeoffs, mean/median/tail latencies, and TTFTs of three methods, using 10 branches per request, on the GSM8K dataset and the R1-Distill-Llama model.

**Complexity-aware scheduling is effective when groundtruth complexities are unavailable at training or test times.** Complexity-aware scheduling with predictions outperforms FCFS on all three datasets, making it applicable when true complexities are unavailable at test time. Moreover, when trained on LLM-labeled complexities (i.e., GSM8K/MMLU datasets), predicted complexities remain effective, demonstrating that our approach does not require provided groundtruth labels for training.

## 4.4 COMPARISON TO COST-AWARE SELF-CONSISTENCY

Past work has been heavily focused on reducing token cost for multi-branch reasoning, while overlooking latency. Incremental Self-Consistency methods (Li et al., 2024; Wan et al., 2025; Aggarwal et al., 2023) generate branches in small batches until reaching a termination condition. While cost-efficient, these methods increase per-request completion time by generating branches sequentially, thereby inflating end-to-end latencies. We compare DUCHESS to ESC: Early-stopping Self Consistency (Li et al., 2024), a representative incremental self-consistency method. Given a request, ESC generates $w$ branches at a time, and terminates the request if all $w$ answers agree (i.e., zero-entropy), or exhausts a predefined total branch budget.

**Incremental Self-Consistency increases latencies and TTFTs.** We set $w = 2$ for ESC with a maximum of 10 branches per request, the most latency-friendly value, which encourages request-level termination. We use 10 branches per request for DUCHESS. Table 2 reports cost-accuracy tradeoffs and serving latencies/TTFTs on GSM8K/R1-Distill-LLama. Although using the fewest tokens among these methods, ESC increases mean latency by 2.7× and 14× compared to Default SC and DUCHESS, respectively. This is caused by 29.1% of requests requiring at least 2 batches of branches and 15.9% requiring 4-5 batches. In contrast, DUCHESS effectively parallelizes branch execution within a request, reducing per-request processing time.

## 4.5 ANALYSIS

**Accuracy of early-terminated CoT branches.** Terminating a CoT early can reduce accuracy if triggered too soon; conversely, it can sometimes improve accuracy by preventing the LLM from overthinking (Hassid et al., 2025). We compared the accuracies of early terminated branches relative

| Component | GSM8K | MMLU | MATH |
|---|---|---|---|
| Prediction | 0.0055 (0.07%) | 0.0022 (0.02%) | 0.0053 (0.02%) |
| Answer Extraction | 0.28 (3.65%) | 0.18 (1.73%) | 0.28 (0.89%) |
| LLM Inference | 7.50 (96.3%) | 10.12 (98.2%) | 30.97 (99.1%) |
| **Total (s/request)** | **7.8 (100%)** | **10.3 (100%)** | **31.3 (100%)** |

Table 3: **DUCHESS adds negligible orchestration overhead.** The average time in seconds that each component of intra-request orchestration spends per request across three datasets, using 10 branches per requst. The percentages represent relative weights per component within total request processing time.

to their uninterrupted accuracies with maximum 4,096 tokens per branch, for 10 branches per request across all three datasets.

Early termination has a negligible impact on GSM8K; only 0.1% of branches that would be correct if uninterrupted are mistakenly early-terminated. This likely reflects GSM8K's lower difficulty. In contrast, 16.4% and 7.8% of branches are mistakenly early-terminated on MMLU and MATH, respectively. However, DUCHESS still matches or exceeds Default SC accuracy because: (1) early termination corrects 20.8% and 22.4% of branches that would otherwise be incorrect (overthinking or incomplete), and (2) majority voting across branches within each request mitigates individual branch errors. One strategy to recover from mistaken early terminations is to query the LLM for multiple answers from an early-terminated branch; if these answers disagree substantially, the branch can be resumed. We plan to explore this multi-shot error recovery mechanism in future work.

**Per-component overhead.** DUCHESS introduces two additional GPU-consuming steps to standard LLM inference at intra-request level: (1) predicting per-branch correctness at each token interval; and (2) extracting final answers from early-terminated branches. Both lie on the critical path, since DUCHESS relies on their output to make decisions at the branch and request levels. They must therefore be lightweight to preserve low end-to-end latency. Table 3 breaks down DUCHESS's mean per-request processing time by component, using 10 branches per request. The overhead of predicting branch correctness is insignificant across three datasets: between $0.02\%$ and $0.07\%$, due to the lightweight predictor design. We additionally report that the predictor's GPU memory footprint is as small as 8.7 to 40.3 MBs, making it feasible to co-locate on the same device as the LLM (hence omitting potential overhead of transmitting activations). The answer extraction overhead is controlled within $0.89\%$ to $3.65\%$, as it costs only 10 tokens per early-terminated branch.

**Ablations and sensitivity analyis.** We provide additional analysis, including (1) ablation of early termination and branch orchestration and (2) DUCHESS sensitivity to selected LLM layers and hyperparameters in Appendix section A.5.

## 5 CONCLUSION AND LIMITATIONS

We have presented DUCHESS, a novel approach that significantly reduces computation and latency for multi-branch LLM reasoning without compromising accuracy. Our key contribution is an intra-request branch orchestration policy. We obtain per-branch correctness probabilities using a lightweight MLP model directly based on LLM activations. The policy dynamically decides an action per branch, including early termination, selective branch-out, and continuation, based on the predictions. The policy terminates the request in its entirety once sufficient final answers are collected to prevent stragglers from inflating completion time. DUCHESS also includes an optional complexity-aware scheduler that prioritizes easier requests, given predicted difficulty levels. We evaluate DUCHESS using three reasoning datasets: GSM8K, MMLU, and MATH, on the R1-Distill-Llama model. At matched accuracy to Default Self-Consistency, DUCHESS reduces computational cost by generating $42\%$-$63\%$ fewer tokens. Using vLLM as the inference engine, DUCHESS reduces mean, P50, and P95 latencies by $52$-$85\%$ and TTFTs by $54$-$99\%$.

**Limitations.** Our intra-request branch orchestration terminates a request based on collected answers. However, these answers have dependencies (e.g., some may come from branches that share the same ancestor before branch-out). Moreover, answers are collected when branches are stopped at varying correctness predictions. We aim to explore answer aggregation and request-termination strategies that account for this information.

# 6 ADDITIONAL INFORMATION

**Ethics statement.** Our study does not raise any ethical concerns.

**Reproducibility statement.** The evaluation datasets (GSM8K, MMLU, MATH, GPQA) and the LLMs (deepseek-ai/DeepSeek-R1-Distill-Llama-8B, internlm/Intern-S1-Mini, and google/Gemma-3-4B-IT) are publicly available on HuggingFace. The codes to replicate the evaluations are provided, although the exact numbers may slightly differ across runs due to randomness in LLM decoding and the selective branch-out process. Our artifacts, including LLM responses, activations, trained predictor weights, and latency measurement logs, are available upon request.

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

# A    APPENDIX

## A.1    PER-BRANCH CORRECTNESS MLP SETTINGS.

**Selection of LLM layers.** We ideally would profile layer-wise accuracy for each dataset and LLM use cases, but are constrained by time and resources. We thus profile the layer-wise accuracy of each model on the GSM8K dataset and identify the optimal layer that maximizes validation accuracy (layer 14, 21, and 17 for R1-Distill-LLama, Intern-S1, and Gemma-3, respectively) to use on the remaining benchmarks.

**MLP general architecture.** The MLP's input layer has a dimension equivalent to the LLM hidden size (4,096 for R1-Distill-Llama and Intern-S1, and 2,560 for Gemma-3), followed by a LayerNorm, then a series of hidden fully-connected (FC) layers, and finally an output layer of dimension 1, corresponding to the logit of probability of correctness.

**Training configurations.** The MLPs are optimized using AdamW with an initial learning rate of $3 \times 10^{-4}$ and 0.1 weight decay. We adjust the learning rate over epochs using StepLR scheduler with step size 5 and gamma 0.8. We use the Binary Cross-Entropy loss function, appropriate for classification tasks. We train the model for up to 15 epochs, and keep the model weights from the epoch with the highest validation accuracy.

**Per-dataset adjustments.** We vary a few MLP architectural and training configurations per dataset and LLM to maximize their validation accuracies. The MLP architectural settings include: the number of FC layers (2-5), the dimension of each FC layer (selected from $[256, 512, 1024, 2048, 4096]$), and the dropout rate on FC layers (0.1-0.3). The training configuration changes include positive class weights (inversely proportional to the fraction of positive labels in training data) and label smoothing (0-0.1). Adjusting these settings specific to each dataset is necessary, as the datasets vary in cardinality and label distributions.

## A.2    DUCHESS HYPERPARAMETER SETTINGS.

**Hyperparameter selection criteria.** We tune DUCHESS hyperparameter based on the cost-accuracy tradeoffs on the validation set. That is, to minimize token usage costs while achieving accuracy that matches or exceeds Default SC. We then use the same settings for latency evaluations. We do not tune hyperparameters based on latency results, as DUCHESS's latency-aware design (i.e., parallel executions, KV reuse from forked branches, straggler preemption) ensures that low token cost leads to low request serving latencies.

Let $c$ and $P$ represent the number of branches per request and a hyperparameter configuration for DUCHESS. Let $Cost(c, P)$ and $Acc(c, P)$ be the cost (token usage) and accuracy of DUCHESS given specific $c$ and $P$ values on the validation set. Let $AccSC(c)$ be the validation accuracy of default self-consistency at $c$ branches per request. Then, the maximum accuracy achieved by Default SC over all possible $c$ values is:

$$A^* = max_{c \in C} AccSC(c)$$

Then, the cost-accuracy tradeoff for hyperparameter values $P$ is given by:

$$OBJ(P, A^*) = \max_{c \in C}(M - Cost(c, P)) \cdot \mathbf{I}[Acc(c, P) \geq A^*]$$

Where $M$ is the upper bound of cost per request, and $\mathbf{I}[Acc(c, P) \geq A^*]$ is an indicator variable with a value of 1 if $Acc(c, P) \geq A^*$, and a value of 0 otherwise. Finally, the optimal parameter choice, $P^*$ is given by:

$$P^* = \text{argmax}_P OBJ(P, A^*)$$

**Auto-tuning.** Because of the well-formulated objective function, DUCHESS's hyperparameter tuning can be effectively automated by optimization algorithms such as grid search and Bayesian opti-

mization (Snoek et al., 2012). In this work, we used a one-parameter-at-a-time grid search to reduce the search space.

**Hyperparamater values.** We detail our hyperparamater choices for using DUCHESS with R1-Distill-Llama and provide insights. These values correspond to the main evaluation results reported in section 4.

$i$ (probing interval): we set $i = 80$ for MMLU and MATH, and a smaller $i = 16$ for GSM8K due to shorter LLM responses.

$\tau$ (early termination threshold): we use the $70^{th}$ percentile of all validation sample predictions for GSM8K, and $80^{th}$ percentile for MATH and MMLU. GSM8K's lower $\tau$ reflects its lower difficulty compared to other datasets, as DUCHESS is able to tolerate more mistakenly early terminated branches while not reducing overall accuracy.

$S$ (number of probes required for early termination): We use $S = 2$ in all cases. We empirically observe that $S = 1$ significantly reduces reasoning accuracy due to over-aggressive early terminations. Setting $S = 2$ with relatively high $\tau$ values (i.e., $70 - 80^{th}$ percentiles of all validation predictions) effectively maintains accuracy.

Other settings: (1) $\lambda$ (sampling temperature for selective branch-out): we use 1 for GSM8K, and 0.8 for MATH and MMLU. We did not test values outside of $0.8 - 1$ ranges. The selected $\lambda$ values keep branch selection weights close or identical to predicted correctness, with slightly increased weight on high-confidence branches for MATH and MMLU, reflecting their advanced difficulties. (2) $\alpha, \beta$ (request-level termination): we use $(0.6, 0.8)$ for $\alpha, \beta$ values on GSM8K and MATH datasets, and $(0.4, 1)$ on MMLU.

### A.3   REQUEST COMPLEXITY PREDICTOR DESIGN AND ACCURACY.

**Predictor architecture.** Our request complexity predictor uses activations from layer 14 of R1-Distill-Llama. We did not test other layers, although they may further improve prediction accuracy and mean latencies. The MLP predictor has an input layer with 4,096 dimension, matching R1-Distill-Llama's hidden sizes, followed by a LayerNorm, then three hidden FC layers with 2048, 1024, 512 dimensions. Each FC layer is followed by a BatchNorm1d, then GeLU activation, and a dropout layer. The dropout rates are 0.15 for first two, and 0.075 for the last FC layer. The output from the last FC layer is passed to another dropout with 0.045 rate, and finally an output layer with dimension 5, corresponding to the number of unique difficulty labels in MATH dataset.

**Training.** The model is optimized using AdamW with a consine annealing learning rate scheduler, and 0.1 label smoothing for regularization. Overall, we applied extensive regularization efforts, but the validation and test accuracies remain around 0.4. Nevertheless, the predicted complexities are sufficient to reduce mean latencies compared to FCFS.

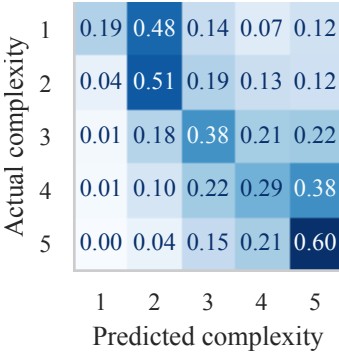

Figure 6: Confusion matrix for predicted request complexities on MATH dataset.

**Confusion matrix for request complexity prediction.** Figure 6 shows the confusion matrix for predicted request complexities on the MATH dataset. Although the predictor has relatively low overall and balanced accuracies (0.42 and 0.39), we observe that it is able to differentiate easier and harder requests well: for example, 81% level-1 requests have predicted complexity levels $\leq 3$, while

81% level-5 requests have predictions $> 3$. Our evaluation demonstrates that these predictions guide DUCHESS to prioritize easier requests and reduce mean latencies.

**LLM-labeled request complexities.** We use online LLM services (i.e., ChatGPT 5.1) to assign complexity labels to GSM8K and MMLU prompts. The LLM-assigned labels are from 1 (easy) to 3 (hard). Overall, DUCHESS's per-request processing time increases with LLM-assigned complexities: 5.5, 7.2, and 10.4 seconds on GSM8K, and 5.4, 9.6, and 16.1 seconds on MMLU, for requests with labels 1, 2, and 3, as assigned by the LLM.

We prompt the LLM as follows for the request complexity labeling task: "You are a grader who evaluates the difficulty of problems for a Large Language Model. Task: assign each problem a difficulty level using ONLY these three labels: 1 = easy: answerable in 1–2 reasoning steps; 2 = medium: requires multiple steps or chaining several facts; 3 = hard: multi-step reasoning, abstraction, or tricky logic. Output format (strict): Return ONLY a CSV with exactly two columns: problem_id, difficulty."

## A.4 EVALUATIONS ON ADDITIONAL LLMs AND DATASETS

We provide additional evaluations to demonstrate the generalizability of DUCHESS across multiple LLMs and datasets.

### A.4.1 ADDITIONAL LLMs.

We evaluate DUCHESS on two additional LLMs: (1) internlm/Intern-S1-Mini (Bai et al., 2025), an 8B reasoning model, and (2) google/Gemma-3-4B-IT (Team et al., 2025), an instruction-tuned, general-purpose model. Results with these LLMs demonstrate DUCHESS generalizes to both reasoning and non-reasoning LLM classes.

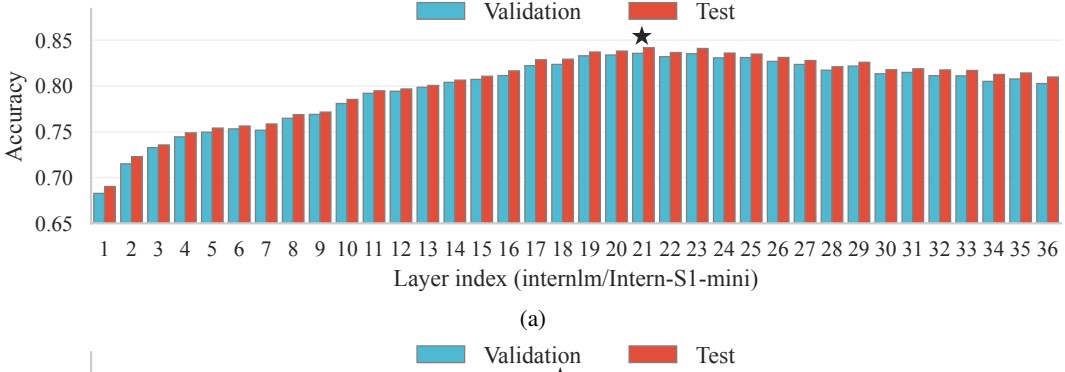

(a)

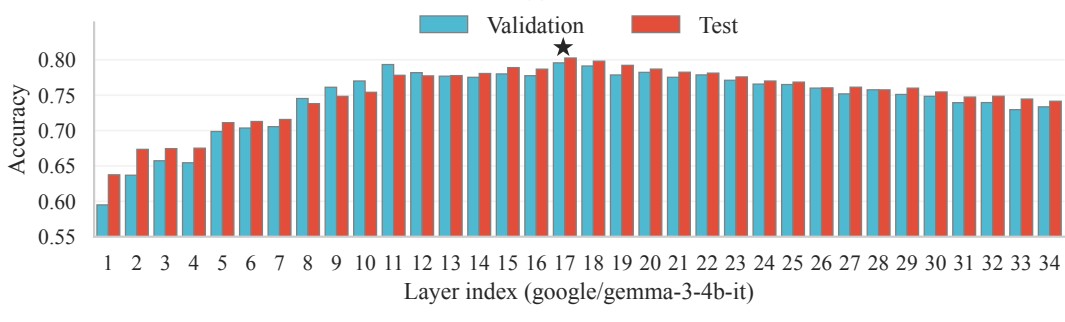

(b)

Figure 7: **Activations produce accurate correctness predictions for reasoning and non-reasoning LLMs.** Per-layer prediction accuracy for branch correctness on the GSM8K dataset, for Intern-S1 (7a) and Gemma-3 (7b). Middle layers yield the highest accuracies for both LLMs, peaking at layer 21 and 17, respectively.

**Layer-wise accuracy profiling.** Figure 7 shows the per-layer probing accuracies on the GSM8K dataset, for Intern-S1 (7a) and Gemma-3 (7b). The highest validation and test accuracies achieved

|  | GSM8K | | MMLU | | MATH | |
|---|---|---|---|---|---|---|
| Method | Cost | Acc. | Cost | Acc. | Cost | Acc. |
| **Intern-S1** | | | | | | |
| Default SC | 3059.9 | 0.925 | 10 026.4 | 0.816 | 7309.0 | 0.702 |
| Dynasor | 3436.2 | **0.928** | 6747.0 | 0.815 | 6960.7 | 0.674 |
| Short-m@k | 2866.5 | 0.916 | 6986.6 | 0.766 | 6430.2 | 0.662 |
| DUCHESS (Ours) | **2508.9** | 0.925 | **4621.0** | **0.821** | **6044.2** | **0.709** |
| **Gemma-3** | | | | | | |
| Default SC | 4006.3 | **0.910** | 3539.4 | 0.658 | 9127.7 | 0.701 |
| Dynasor | 3525.2 | 0.900 | 3116.5 | 0.640 | 7568.0 | 0.663 |
| Short-m@k | 3600.8 | **0.910** | 3357.0 | 0.660 | 8330.4 | 0.701 |
| DUCHESS (Ours) | **3136.8** | **0.910** | **2863.7** | **0.667** | **7829.8** | **0.704** |

Table 4: **DUCHESS reduces computational cost without compromising accuracy for two LLMs.** Cost–accuracy tradeoffs on GSM8K, MMLU, and MATH using Intern-S1 and Gemma-3, using 10 branches per request. Results are averaged from 10 trials with errors omitted.

by both models match those reported for R1-Distill-LLama in Figure 2, demonstrating the generalizability and consistency of our activation-based prediction technique across broad LLM classes. We use the identified layers on GSM8K for other datasets due to time and resource constraints. Ideally, we would perform per-layer accuracy profiling for every dataset and LLM use case.

**Cost-accuracy tradeoffs.** We report the cost (i.e., average tokens per request) and accuracy achieved at 10 branches per request. In the future, we would like to evaluate the full cost-accuracy Pareto-frontier. We prioritize evaluation at 10 branches to match settings for latency measurements.

Table 4 reports the average token cost per request and accuracy on GSM8K, MMLU, and MATH datasets, for Intern-S1 (top) and Gemma-3 (bottom) models. Overall, DUCHESS consistently reduces cost while matching or improving Default SC's accuracy, demonstrating generalizability across diverse use cases.

Both Intern-S1 and Gemma-3 generate significantly shorter responses than R1-Distill-Llama (e.g., at 10 branches, Gemma-3 takes 3136.8 tokens per request on GSM8K, while R1-Distill-Llama takes 7763.4 tokens per request, as shown in Figure 3 in the submission). This may be due to R1-Distill-Llama naturally outputs its internal "thinking" processes, while Intern-S1 and Gemma-3 require COT-styled prompting. Shorter responses reduce room for token pruning; nevertheless, DUCHESS still achieves sizable cost reductions.

Dynasor increases token cost compared to Default SC for the Intern-S1 and GSM8K setting. Dynasor frequently queries the LLM for intermediate answers, which costs 10 additional tokens per 64 decoding steps. For shorter responses, frequent answer extractions inflate token usage and outweigh early termination benefits.

**Latencies and TTFTs at default request rates.** Similar to section 4, we randomly generate request schedules following Poisson distributions. The mean request arrival rate for each ¡dataset, LLM¿ setting is based on the upper quantile of Default SC's per-request processing time: for Intern-S1, the average queries-per-minute (QPM) is 10, 2, and 4 for GSM8K, MMLU, and MATH, corresponding to the upper quantiles of 6.2, 32.1, and 16.8 seconds per request; for Gemma-3, the average QPMs are 10, 15, and 3, based on upper quantiles of 6.7, 4.7, and 19.6 seconds per request, respectively. We upgrade vLLM to 0.11.0 as requested by Intern-S1.

Overall, DUCHESS consistently reduces latencies and TTFT values for all tested datasets and LLMs, demonstrating generalizability and practicability across diverse use cases. Short-m@k achieves the lowest tail latencies and all TTFTs on MMLU/Intern-S1 setting; however, it reduces accuracy from Default SC by a large margin shown in Table 4 (-0.05). In contrast, DUCHESS slightly improves accuracy (+0.005) while reducing latency and TTFT values.

**Latencies as functions of request rates.** Figures 8 and 9 show latencies as function of QPMs for Intern-S1 and Gemma-3, respectively. The results are similar to those reported on R1-Distill-LLama (Figure 4): at matched latency target to Default SC's performance under normal loads, DUCHESS

| Method | GSM8K | | | MMLU | | | MATH | | |
|---|---|---|---|---|---|---|---|---|---|
| | **Mean** | **P50** | **P95** | **Mean** | **P50** | **P95** | **Mean** | **P50** | **P95** |
| **Intern-S1 Latency (s)** | | | | | | | | | |
| Default SC | 38.2 | 31.9 | 85.1 | 75.4 | 58.7 | 198.8 | 263.2 | 267.6 | 459.1 |
| Short-m@k | 11.5 | 8.7 | 31.6 | 15.4 | 10.1 | **46.5** | 37.5 | 17.7 | 130.3 |
| DUCHESS | **9.3** | **7.2** | **24.2** | **14.9** | **7.8** | 58.8 | **29.5** | **13.2** | **98.1** |
| **Intern-S1 TTFT (s)** | | | | | | | | | |
| Default SC | 32.8 | 26.6 | 79.2 | 53.1 | 35.1 | 170.4 | 246.3 | 248.2 | 433.3 |
| Short-m@k | 7.2 | 4.2 | 27.4 | **4.9** | **0.1** | **27.0** | 26.7 | 6.4 | 117.1 |
| DUCHESS | **5.3** | **3.1** | **19.8** | 5.9 | **0.1** | 39.1 | **20.1** | **3.0** | **87.6** |
| **Gemma-3 Latency (s)** | | | | | | | | | |
| Default SC | 136.2 | 139.8 | 247.9 | 302.7 | 283.1 | 495.7 | 36.5 | 28.1 | 100.2 |
| Short-m@k | 12.2 | 8.6 | 37.1 | 24.8 | 23.0 | 71.4 | 17.9 | 14.2 | 48.8 |
| DUCHESS | **8.8** | **5.5** | **28.9** | **14.2** | **11.7** | **35.5** | **14.4** | **11.3** | **37.4** |
| **Gemma-3 TTFT (s)** | | | | | | | | | |
| Default SC | 130.1 | 132.0 | 241.8 | 298.2 | 278.6 | 492.0 | 22.8 | 12.7 | 82.3 |
| Short-m@k | 8.1 | 4.0 | 29.9 | 21.1 | 18.8 | 67.4 | 7.8 | 0.1 | 36.1 |
| DUCHESS | **5.3** | **1.6** | **24.0** | **10.8** | **8.1** | **32.7** | **5.5** | **0.04** | **24.6** |

Table 5: **DUCHESS reduces request latecies and TTFTs on additional LLMs.** Mean/P50/P95 latencies and time-to-first-token (TTFT) in seconds, measured from one run using 10 branches per request. Request arrival follows random Poisson distributions.

| Method | GPQA | |
|---|---|---|
| | **Cost** | **Acc.** |
| Default SC | 6970 | 0.28 |
| Dynasor | 6322 | 0.29 |
| Short-m@k | 4863 | 0.30 |
| DUCHESS (Ours) | **4482** | **0.31** |

Table 6: **DUCHESS reduces computational cost without compromising accuracy on GPQA.** Average token cost per request and accuracy on GPQA at 10 branches per request, using R1-Distill-LLama. Results are averaged from 10 trials with errors omitted.

| Method | GPQA | | |
|---|---|---|---|
| | **Mean** | **P50** | **P95** |
| **Latency (s)** | | | |
| Default SC | 36.4 | 22.0 | 124.4 |
| Short-m@k | 12.4 | 8.7 | 33.8 |
| DUCHESS (Ours) | **10.9** | **7.6** | **30.8** |
| **TTFT (s)** | | | |
| Default SC | 20.9 | 5.3 | 107.0 |
| Short-m@k | 3.4 | **0.04** | 23.7 |
| DUCHESS (Ours) | **3.3** | **0.04** | **18.7** |

Table 7: **DUCHESS reduces request latecies and TTFTs on GPQA.** Mean/P50/P95 latencies and time-to-first-token (TTFT) in seconds, measured from one run using 10 branches per request.

can process approximately 1.4-3× more requests on average, demonstrating robustness to bursts of requests across LLM classes.

### A.4.2 ADDITIONAL DATASET.

We include GPQA (Rein et al., 2024) as an additional benchmark. We report the cost-accuracy tradeoffs and latency/TTFT measurements only for R1-Distill-Llama at 10 branches per request due to time and resource constraints. We use layer 14's activation as input for correctness predictions.

**Cost-accuracy tradeoffs.** Table 6 reports the average token cost and accuracy for all methods. Default SC achieves low accuracy (0.28) on the GPQA dataset. DeepSeek officially reports 0.49 accuracy (Guo et al., 2025) on the Diamond subset, using 64 branches and a maximum of 32,768 tokens per branch. Our accuracy gap most likely comes from fewer branches with shorter lengths. We plan to extend our branch and max token settings to match DeepSeek's settings as future work. Overthinking contributes to a significant portion of the mistakes: among all branches with incorrect final answers, 40.7% had at least one correct intermediate answer but switched in the end.

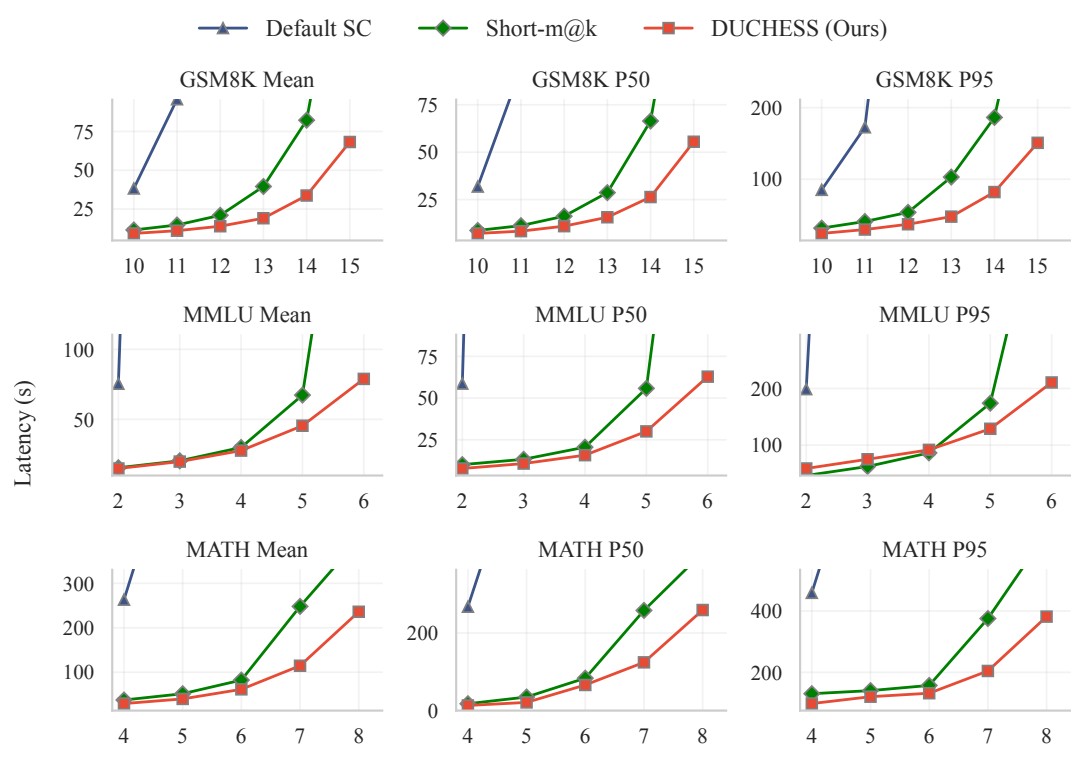

Figure 8: **DUCHESS is robust to bursts of requests for Intern-S1.** Mean, P50, and P95 latencies as a function of request arrival rate (10 branches per request).

DUCHESS achieves the highest accuracy, with 0.03 points (approximately 11%) higher than the default SC. This demonstrates DUCHESS can successfully prevent LLM overthinking by combining (1) branch early terminations and (1) expanding "promising" branches with higher predictions. DUCHESS has the lowest token cost among the techniques, with approximately a 36% reduction from the Default SC.

**Latencies and TTFTs at default request rates.** We randomly generate a request arrival schedule following a poisson distribution with an average of 3 QPM (based on the upper quantile of Default SC's average processing time per request: 18 seconds). We include short-m for reference, although the accuracy is unmatched. Overall, DUCHESS achieves 70%, 65%, 75% reduction in the mean, median, and tail latencies, along with 84%, 99%, 82% reductions in the mean, median, and tail TTFTs. The reductions are similar to those of other evaluated datasets (Table 1), demonstrating practical serving latency reductions across diverse tasks. We omit latencies under increasing QPM values.

## A.5    ABLATION AND SENSITIVITY ANALYSIS

**Ablating early termination and branch orchestration.** We add DUCHESS-ET, which only performs prediction-based early terminations of branches without cross-branch orchestration (i.e., selective branch-out, request-level termination). Figure 10 reports the cost-accuracy frontier with the number of branches per request varying from 5 to 10, on GSM8K/R1-Distill-Llama. DUCHESS-ET is more effective at fewer branches per request. However, as the number of branches increases, expanding on the more "promising" branches can lead to a faster majority with high accuracy. At 10 branches per request (the right-most marker on each curve), DUCHESS-ET alone reduces tokens by 30% compared to default self-consistency, and adding branch orchestration reduces them by an additional 12.3%.

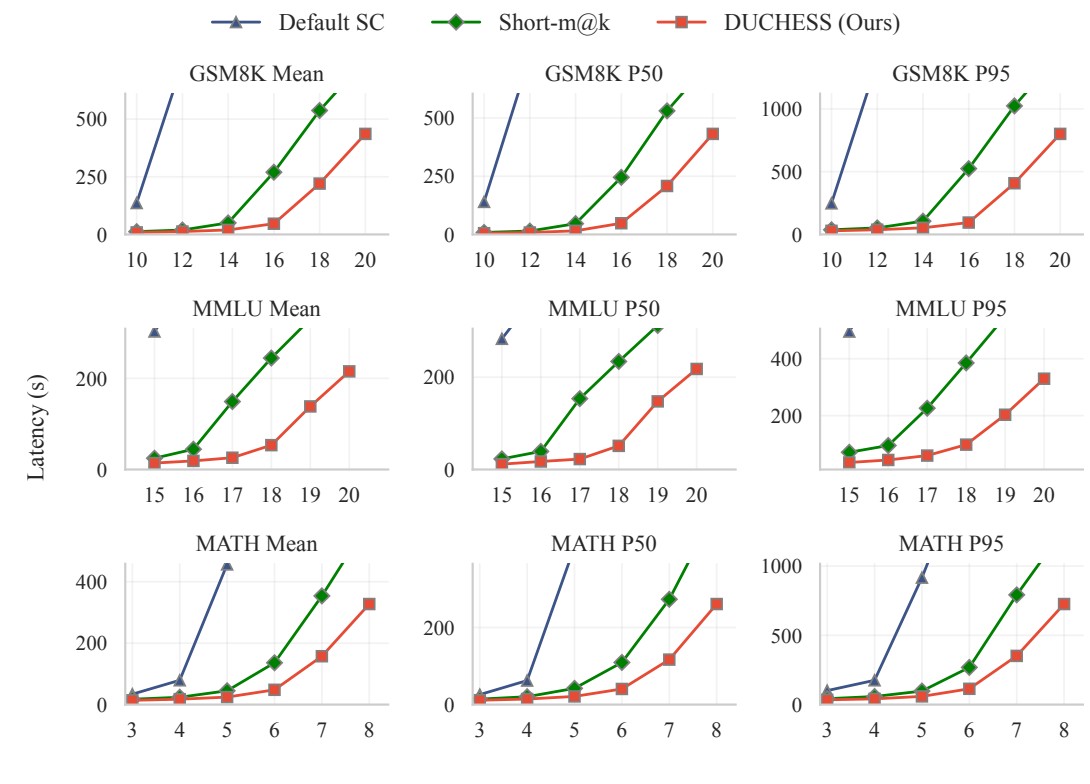

Figure 9: **DUCHESS is robust to bursts of requests for Gemma-3.** Mean, P50, and P95 latencies as a function of request arrival rate (10 branches per request).

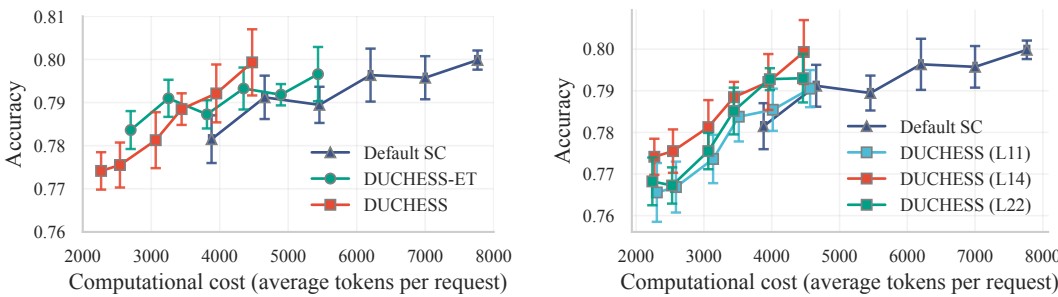

Figure 10: **Branch orchestration yields better cost-accuracy tradeoffs at more branches.** Cost-accuracy tradeoffs for DUCHESS with only early termination, and with branch orchestration added, on GSM8K/R1-Distill-Llama.

Figure 11: **DUCHESS is not highly sensitive to the selected LLM layer.** Cost-accuracy tradeoffs for DUCHESS using the 11th, 14th (best), and 22th layer of R1-Distill-Llama for prediction, on the GSM8K dataset.

**Sensitivity to selected LLM layers.** Middle-range layers consistently yield the most accurate predictions for branch correctness in three LLMs (Figures 2, 7a, 7b). The gaps among middle-range layers are small. Therefore, our method is robust to the specific layer choice, as long as the layer yields sufficiently accurate predictions. Figure 11 reports the cost-accuracy tradeoffs of DUCHESS on GSM8K, using predictions derived from three R1-Distill-Llama layers: 11, 14 (best), and 22. Layers 11 and 22 are the left and right boundaries of the LLM middle region. Overall, L11 and L22 perform closely to L14 in terms of token usage. The accuracy gap is at most 0.01, which could be reduced by optimizing hyperparameters specific to each layer choice.

**Sensitivity to DUCHESS hyperparameters.** We show how the cost-accuracy tradeoff of DUCHESS changes with three key hyperparameters: $\tau$ (early termination threshold), $i$ (probing interval), and $S$ (number of probes with probability $\geq \tau$ required for early termination). Figure 12

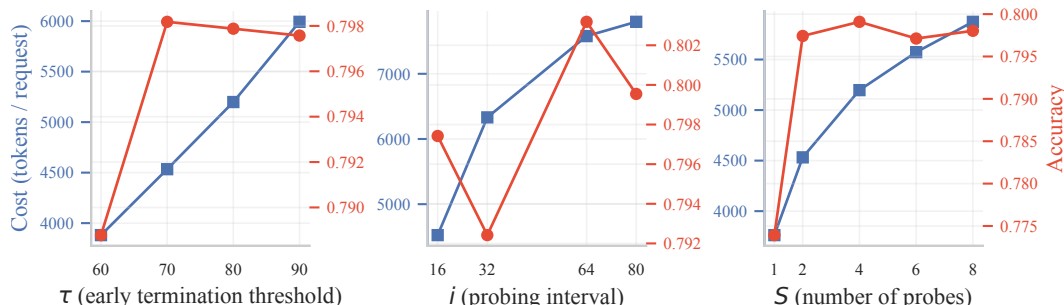

Figure 12: **Sensitivity to DUCHESS hyperparameters.** The average token cost per request (left y-axis) and accuracy (right y-axis) as functions of DUCHESS hyperparameter values. Results are reported on the GSM8K dataset using the R1-Distill-Llama model. $\tau$ values are percentiles of all validation-set predictions. The default values are 70/16/2 for $\tau/i/S$, respectively. Results are averaged from 5 runs with errors omitted.

plots the cost (shown in blue color on the left y-axis) and accuracy (red color on the right y-axis) changes as functions of different hyperparameter values. Results are reported on GSM8K/R1-Distill-Llama at 10 branches per request. For each subplot, we vary one hyperparameter (marked in the x-axis) while keeping others to their selected values: 70/16/2 for $\tau/i/S$ that give optimal validation cost-accuracy tradeoffs, as discussed in section A.2.

For all three hyperparameters, smaller values reduce token cost per request. This aligns with expectation, since smaller $\tau$ and $S$ values lower the early-termination requirements, whereas smaller $i$ values increase probing frequency. Overly small $\tau$ and $S$ values reduce accuracy by up to $0.024$. However, overly large values only inflate token usage without accuracy gains. Accuracy varies non-monotonically with respect to $i$, as overly frequent probing may aggressively terminate branches and cause accuracy drops, while probing too rarely reduces the chances of preventing CoT overthinkings.

A.6    THE USE OF LARGE LANGUAGE MODELS (LLMS)

We use LLMs to (1) condense writings and fix grammatical errors, (2) help identify related works that may have been missed, and (3) format figures and tables. We additionally use LLM-integrated Integrated Development Environments (IDEs) to assist programming.

