# OpenReview forum: "Intra-Request Branch Orchestration for Efficient LLM Reasoning"
_ICLR.cc/2026/Conference — Submitted to ICLR 2026_

### Official Review · Reviewer_XiJK · 2025-10-31

**Soundness:** 4
**Presentation:** 4
**Contribution:** 4
**Rating:** 8
**Confidence:** 3

**Summary:**

The paper introduces an innovative system called DUCHESS (Intra-request branch orchestration for efficient LLM reasoning). It proposes to make Large Language Models (LLMs) faster and cheaper to run, especially when they are solving complex problems using multi-branch reasoning methods (like Self-Consistency or Chain-of-Thought). While these reasoning algorithms help LLMs achieve higher accuracy on difficult tasks, they typically increase the number of output tokens generated, leading to higher computational costs and longer waiting times (latency) for users. The proposed system DUCHESS tackles this challenge by intelligently managing these reasoning paths, or "branches," to maintain accuracy while drastically reducing cost and latency.
The work is primarily based on the following two mechanisms:
1. Intelligent Intra-Request Branch Orchestration (Stopping, Forking, Continuing).
2. Optional Inter-Request Complexity-Aware Scheduling.

*Trying to make the review bullet points for the clarity and easy to respond.

**Strengths:**

The paper provides strong evidence that the fundamental approach of DUCHESS is sound and highly effective, leading to significant, quantifiable improvements over existing baselines:
1. Dominance on the Cost–Accuracy Pareto Frontier.
2. Efficacy of Lightweight Prediction and Minimal Overhead.
3. Substantial Latency and Straggler Reduction.
4. Optional Complexity-Aware Scheduling.
5. Novel Intra-Request Branch Orchestration Policy.

**Weaknesses:**

The paper is transparent in listing several limitations, particularly regarding the aggregation of results and the scope of implementation:
1. Dependencies in Answer Aggregation.
2. Limited Scope of Complexity-Aware Scheduling.
3. Predictor Layer Constraints.
4. Low Complexity Predictor Accuracy.
5. Hyperparameter Tuning Range.
6. Scheduling Overhead.

**Questions:**

A couple of questions here that the paper clearly covers as limitations-
1. About the Enhanced Answer Aggregation and Termination Strategy: I would wish for request termination strategies that explicitly account for dependencies among collected answers.
2. Broader Applicability and Optimization of Inter-Request Scheduling:
 a. Extension of complexity-aware scheduling to datasets that lack inherent difficulty labels.
 b. Improvement in the accuracy of the request complexity predictor.
 c. Profiling of alternative LLM layers.
 d. Mitigation of queueing delays caused by prefilling.
3. More Extensive Predictor Tuning and Exploration.

---

> ### Author Response · Authors · 2025-11-22
> **Author Response**
>
> We thank the reviewer for the support of our work. We appreciate the reviewer acknowledging our key decision choices.
>
> **Q1: Request termination strategies accounting for dependencies.**
>
> We thank the reviewer for highlighting this point. A natural way to account for dependencies among collected answers is to introduce a **weighted voting mechanism**. Initially, all branches share the same weight. At each branch-out, a branch’s weight is redistributed among its descendants. When a branch produces an answer via early termination, we record its current weight and incorporate it into the final vote, reducing bias caused by duplicated or closely related branches. As noted by the reviewer, we briefly mentioned this idea in the limitations section.
>
> We plan to explore this direction in future work, focusing on: (1) how branch weight should decay after each branch-out, and (2) how to design a weighted voting procedure that integrates both branch weights and predicted correctness probabilities for early-terminated answers.
>
> **Q2a: Extension of complexity-aware scheduling to datasets without provided difficulty labels.**
>
> We appreciate the reviewer’s question. For datasets that do not provide inherent difficulty labels, the complexity-aware scheduler can still be applied by predicting difficulty directly from the prompt. In practice, this can be done by training a lightweight model or probe to estimate request complexity (e.g., predicted correctness probability, expected reasoning depth, or expected branch divergence) from the prompt representation. These predictions can then be used as inputs to the scheduler in the same way as ground-truth difficulty labels.
>
> We are currently experimenting with prompt-based complexity prediction and will update the rebuttal with results as they become available.
>
> **Q2b: Improving the accuracy of the request complexity predictor.**
>
> We agree that improving the accuracy of the request-complexity predictor is important. Beyond the current probe, we are experimenting with richer prompt representations (e.g., multi-layer embeddings, token-level features) and lightweight calibration models to refine the predicted correctness probability. We will update the rebuttal with results once these experiments complete.
>
> **Q2c: Profiling of alternative LLM layers.**
>
> We observe that DUCHESS is not highly sensitive to the specific layer choice; i.e., any middle-range layer that produces reasonable correctness prediction accuracy can effectively reduce reasoning costs and latencies while maintaining accuracy. We have provided additional evaluation results with multiple LLM layers with details in section 2.3 of the general response.
>
> **Q2d: Mitigation of prefilling-caused queuing delays.**
>
> Prefilling-caused queueing delay exists since our request complexity predictor requires prefilling results (i.e., activation of the first decoded token) as the input. To avoid such delays, we could modify the request complexity predictor to use alternate inputs.
>
> **Q3. More extensive predictor tuning and exploration.**
>
> We explored several approaches for implementing the predictor during DUCHESS development. One alternative we tested was predicting correctness using activations from the last few decoded tokens, processed by a small transformer or sequence model (e.g., an LSTM). However, we found that a simple MLP operating on the last token’s activation performs best in both accuracy and efficiency.
>
> Going forward, we plan to explore LLM self-assessed correctness predictions, which may further improve accuracy and remove the need for training a separate MLP.

---

### Official Review · Reviewer_GqzD · 2025-11-01

**Soundness:** 2
**Presentation:** 2
**Contribution:** 2
**Rating:** 2
**Confidence:** 4

**Summary:**

The paper introduces DUCHESS, an LLM serving system that aims to reduce computational cost and latency in multi-branch reasoning setups. The system incorporates a linear probing model that leverages the transformer model's activations to predict correctness of a reasoning branch. These predictions are utilized by the system to perform branch level operations including 1) early stopping, 2) additional branching, and 3) continued generation. On datasets like GSM8K, MATH, and MMLU, the work demonstrates token length reductions of 42%-63%, and 57%-81% reduction in mean latency in concurrent request serving scenarios.

**Strengths:**

- The core intuition and methodology are well-grounded in observations from prior research work about:
    - Informativeness of the middle layers of transformer models.
    - Linear probing techniques for extracting signals from prior layers.
    - However, concerns exist regarding hyperparameter selection and generalizability (especially with regards to model architecture families).
- Overall, the paper is well structured with informative diagrams/tables, along with system workflow explanation.The writing is mostly succinct with minimal filler.
- The work showcases results with deployment considerations in mind. Specifically, the throughput numbers (queries per minute) in figure 4, showcase promising results for production environments.
- Additional component overhead (branch correctness prediction/answer extraction), in terms of latency, provide a confident argument in favor of the added complexity.

**Weaknesses:**

- The connection between the two main contributions, i.e., 1) intra-request orchestration and 2) inter-request scheduling, feels disconnected.
- Explanations about the choice of hyperparameter selection is incomplete at times. For example: It is mentioned in line 211 that probing for branch correctness is performed at intervals of $i=16$ tokens, while in line 215, it is mentioned that a suitable range is $16 \leq i \leq 80$.
- The originality seems somewhat incremental:
    - The core idea of early termination based on hidden states has been explored (e.g., Zhang et al., 2025; Afzal et al., 2025).
    - The probing technique, although claimed to be sufficiently advanced compared to prior works (see lines 127-135), does not seem entirely novel.
    - The complexity-aware scheduling component feels like an add-on rather than a core contribution, and is only evaluated on one dataset (see figure 5).
- The experiments are performed with just one model, i.e., DeepSeek-R1-Distill-Llama-8B. It is unclear if the methods proposed generalize across different models and if the technique applies to non-reasoning models
- In lines 204-207, it is said that, based on the validation performance of the model on GSM8K, the layer 14 was chosen for correctness prediction. Selection of the prediction layer based on empirical results on a single dataset/model combination raises concerns about applicability in real-world deployment scenarios.
- It is unclear whether the methods generalize to tasks involving longer reasoning traces (GPQA-Diamond, for example), as opposed to the presently evaluated datasets (GSM8K, MATH, MMLU), which require relatively lower generation lengths.
- Hyperparameter choices such as the probing interval ($i=16$), or the branch termination threshold($\tau$) would benefit from further analysis and/or advice about selection criteria in deployment scenarios where single-dataset validation data might not suffice.
- In lines 148-150, it is mentioned that the termination of entire request, once a "sufficient" number of answers is collected, prevents stragglers from delaying completion. The claim would benefit from further explanation of the so-called "straggler" branches and how the method prevents them. The main concern here is whether the remaining long-running branches pre-empted by the method, are actually wasteful in practice, or they meaningfully contribute towards alternate reasoning path expansion.
- The work motivates the proposed method from the perspective of improving multi-branch reasoning. With this regard, comparison with existing techniques which already attempt to tackle this problem, such as ESC, RASC, Adaptive Consistency would better support the claims.

The work's significance is primarily in its engineering integration and empirical validation rather than algorithmic breakthrough.

**Questions:**

- Q1. Regarding linear probing models for branch correctness prediction:
(Refer Appendix A.1, lines 710-711) Why are the hidden layer sizes of the MLP different for MMLU vs. GSM8K/MATH datasets?
- Q2. Regarding Experimental Settings:Why is the max token length per branch is capped at 4,096 (see lines 304-305) ?

---

> ### Author Response · Authors · 2025-11-22
> **Author Response (Part 1/2)**
>
> We thank the reviewer for the thoughtful feedback and valuable suggestions, which helped us improve the clarity and quality of our work. We also thank the reviewer for recognizing the clarity and succinctness of our writing.
>
> **W1: Intra-request orchestration and inter-request scheduling feel disconnected.**
>
> The two components, i.e., intra-request branch orchestration and inter-request complexity-aware scheduling, share the common goal of reducing request serving latency. Latency is dependent on two major factors: (1) queueing time, and (2) request processing time. The inter-request scheduling component reduces queueing, while the intra-request orchestration shortens per-request processing time. While these techniques work relatively independently of each other, together they form a comprehensive, end-to-end solution for reducing latency.
>
> **W2: Explanation of hyperparameter choices is incomplete at times.**
>
> We detailed MLP architecture configuration and DUCHESS hyperparameter selection process in sections 3.1 and 3.2 of the general response, respectively.
>
> We thank the reviewer for pointing out the inconsistent description of $i$. In the submission, we have used $i$ to refer to two things: (1) token interval of collecting activations from training prompts (line 196), which is 16 tokens for all three datasets, and (2) token interval to probe correctness at inference time, which varies between 16-80 on a per-dataset basis (line 215). We will resolve this by using different symbols.
>
> **W3 (bullet point 1 & 2): The originality seems incremental, e.g., both early termination based on hidden states and probing.**
>
> We thank the reviewer for pointing this out, and we would like to clarify our overall framing.
> - **Goal.** Our goal is to reduce both serving latency and token cost in branch-based reasoning.
> - **Why.** In real LLM serving workloads, latency—not just token count—is the dominant constraint. Techniques that reduce tokens but ignore system behavior (e.g., parallelism limits, KV-cache reuse, batching effects) often fail to translate into actual latency improvements.
> - **What.** To address this, we propose an intra-request branch orchestration framework that manages three actions—branch-out, continuation, and early termination—while explicitly accounting for system-level factors that affect latency.
> - **Difference from prior work.** Prior approaches (e.g., Zhang et al.) focus mainly on cost reduction. Our method is latency-aware, system-oriented, and uses probing-based early termination as only one component of a broader orchestration mechanism.
>
> **W3 (bullet point 3): The complexity-aware scheduling feels like an “add-on” and is only evaluated on one dataset.**
>
> We appreciate the reviewer's question. We extended the complexity-aware scheduling to all three datasets, using LLM-generated complexity labels as training groundtruths for GSM8K and MMLU.
>
> [Figure: comparison of compexity-aware scheduling to First-Come-First-Served on three datasets](https://i.ibb.co/vxktYXZ3/latency-dsk-all-scheduling.png)
>
> Complexity-aware scheduling outperforms FCFS with predicted complexities, and with predictor trained on LLM-labeled groundtruths.
>
> **W4: Experiments are done on one LLM only; unsure about generalizability to other reasoning or non-reasoning LLMs.**
>
> We added one additional reasoning LLM (Intern-S1-mini) and a non-reasoning LLM (Gemma-3-4B-IT) and evaluated the cost/accuracy tradeoffs and serving latencies/TTFTs on three datasets: GSM8K, MMLU, MATH. Please refer to the general response section 1.1 for detailed results. Overall, our results demonstrate DUCHSS’s generalizability across broad model classes.
>
> **W5: Selection of layer 14 is based on a single dataset/model and may raise concerns in real-world deployments.**
>
> We appreciate the reviewer’s concern, and we would like to clarify our rationale for layer selection. It is ideal to profile all layers for all dataset/model use cases. In line 722-723 of the original submission, we stated that we did not perform the profiling on other datasets due to time and resource constraints. We provide two additional insights regarding layer selection:
> - DUCHESS is not highly sensitive to the specific layer choice; i.e., any middle-range layer that produces reasonable correctness prediction accuracy can effectively reduce reasoning costs and latencies while maintaining accuracy. We provided cost-accuracy evaluation using alternative layers in general response section 2.3.
> - Middle-range layers consistently yield the highest prediction accuracies across multiple LLMs tested, as shown in the section 1.1.1 of general response.
>
> **W6: Unclear of generalizability to tasks involving longer traces.**
>
> We have added GPQA as an additional benchmark. The cost-accuracy tradeoffs and latecy/TTFT results are reported in section 1.2 of the general respose. Our results demonstrate the generalizability of DUCHESS to more complex tasks.

---

> ### Author Response · Authors · 2025-11-22
> **Author Response (Part 2/2)**
>
> **W7: Hyperparameter selection criteria in production deployment scenarios where single-data validation may not suffice.**
>
> We appreciate the reviewer’s interest in practical, production LLM deployment. For the single-dataset use case, we have detailed our hyperparameter optimization method in Section 3.2 of the rebuttal. For more complex use cases, we advise raising the bar of early termination: e.g., high $\tau$ and $S$ values, to ensure reasoning accuracy.
>
> **W8: Explain “straggler” branches, and if they are wasteful in practice or contribute towards reasoning path expansion.**
>
> “Stragglers” refer to the long-running branches that remain unfinished when answers from the other early or naturally terminated branches reach a clear consensus. In such settings, the additional decoding is wasteful as its unlikely to change the final consensus of the other branches.
>
> We run DUCHESS at 10 branches per request using R1-Distill-Llama model and GSM8K dataset, but with one alternation: at each request termination, we allow the still-running branches (i.e., the "stragglers") to finish. We account for the additional tokens and final answers from those stragglers when computing final cost-accuracy tradeoffs. Finishing the stragger branches changed the end majority on only 4.8\% of requests in both directions (i.e., from correct to incorrect, and vice versa). However, the average tokens per request increases from 4433.2 to 7950.5 (+79\%). These results indicate that stragglers rarely improve answer quality but significantly increase decoding cost, supporting our design choice to terminate the request early once sufficient consensus is reached.
>
> **W9: Comparison to existing methods that focus on improving multi-branch reasoning**
>
> We have added a baseline, Early Self-Consisteny (ESC) as suggested by the reviewer. The cost-accuracy tradeoff and latency/TTFT measurements are in section 1.3 of the general response. Overall, ESC focuses on reducing tokens by generating branches incrementally. However, this increases per-response completion time and inflates mean latency by 2.7X and 14.1X compared to Default SC and DUCHESS, on GSM8K and R1-Distill-Llama use cases. We plan to extend ESC to more datasets, and add another baseline method, Slim-SC, in the camera-ready version.
>
> **Q1: Why do the MLP hidden layer sizes differ on MMLU vs. GSM8K/MATH?**
>
> While MLP input (i.e., LLM activation) and output (i.e., logit for correctness probability) layers are consistent across datasets, we tune MLP architectures (e.g., number of hidden layers, hidden sizes, drop-out rates) and training configurations (e.g., learning rates, positive class weights) per dataset. This is because the three datasets differ in cardinality and label distributions. Please check section 3.1 of the general response for a detailed discussion.
>
> **Q2: Why is the max token length per branch capped at 4,096?**
>
> A max length of 4,096 per response is sufficient for many reasoning tasks. We would like this value to be sufficiently large so that most branches finish (otherwise, they get chopped at the end without producing an answer).
>
> We empirically found 4,096 is sufficient for the use cases evaluated in the submission. At 10 branches per request, the maximum possible tokens per request is 40,960. In Figure 2 of the original submission, the average tokens per request of Default SC are 7763.4, 9598.3, 25390.5 for GSM8K, MMLU, and MATH, which are all far below the maximum 40,960. More complex tasks, such as GPQA, may require longer CoT traces and more branches.

---

### Official Review · Reviewer_cMbD · 2025-11-01

**Soundness:** 3
**Presentation:** 3
**Contribution:** 3
**Rating:** 4
**Confidence:** 2

**Summary:**

The paper introduces DUCHESS, a serving‑time controller for multi‑branch chain‑of‑thought (CoT) inference. A lightweight MLP probe on intermediate activations predicts per‑branch correctness, enabling three actions at decode time: early‑terminate, selectively fork promising branches (reusing KV cache), or continue. A request terminates when a consensus criterion is met. In multi‑request settings, an optional scheduler prioritizes prompts predicted to be easier. Probes are trained using periodic CoT probes (interval i) with mid‑layers performing best; key results report 42–63% token reductions at iso‑accuracy and large mean/tail latency drops under vLLM, further improved by difficulty‑aware scheduling.

**Strengths:**

Beyond prior early-stopping work in reasoning (e.g., DeepConf), this paper treats early stop as an end-to-end serving problem with a hardware/system-aware design: a lightweight MLP activation probe that runs co-resident with the model to avoid activation transfer, KV-cache-aware selective branching, and request-level consensus with optional SJF-style difficulty scheduling. This coupling yields consistent iso-accuracy token savings (≈42–63%) and sizable wall-clock gains—including mean/P95 latency and TTFT—on vLLM under load.

Consistent token and latency savings at matched accuracy; helpful layer‑wise probe analysis (mid‑layer peak).

**Weaknesses:**

- Probe seems trained per dataset/model; multi-task/domain robustness is not stress‑tested.
- Comparisons to incremental/entropy‑aware branching (e.g., ESC/RASC) and similarity‑pruning (Slim‑SC) would better calibrate gains.
- The paper reports per‑component costs, but explicit wall‑clock accounting under large batches/multi‑GPU and periodic probing vs. the achieved savings would help.
- Difficulty‑aware prioritization may starve "hard" requests at load; fairness/quality safeguards are not fully analyzed.

**Questions:**

- Sensitivity to 𝑖 (probing interval), correctness threshold 𝜏, and the consecutive‑round parameter S (for early termination). Any auto‑tuning procedure?
- Does a probe trained on one backbone/size transfer to others, or is per‑model retraining required?
- Ablate early termination vs. branch‑level pruning: what fraction of latency/tokens each contributes?
- Under difficulty‑aware scheduling, how are starvation or fairness handled when many "hard" jobs arrive?

---

> ### Author Response · Authors · 2025-11-22
> **Author Response**
>
> We thank the reviewer for acknowledging our efforts to build a practical, low-latency system for serving reasoning requests and for the constructive feedback that helps us strengthen our evaluations.
>
> **W1: Probe trained per dataset/model; multi-task/domain robustness is not evaluated.**
>
> We appreciate the reviewer’s observation. Our current setting is to train one MLP probe per dataset/LLM setting. For the camera-ready, we will provide cross-dataset transferability analysis to evaluate the probe’s robustness across tasks and domains.
>
> **W2: Comparison to incremental/entropy-aware branching and similarity pruning**
>
> We thank the reviewer for this suggestion. We have added Early Self-Consistency (ESC) as an additional baseline method. Please check section 1.3 of the general response. We observe that ESC delivers better cost/accuracy trade-offs but significantly increases latency: on GSM8K dataset and R1-Distill-Llama model, ESC increases mean latency by 2.7X and 14.1X under normal request loads, compared to Default SC and DUCHESS, respectively. This is because ESC generates branches incrementally, whereas DUCHESS’s latency-aware design ensures fully parallelizable branch execution. We plan to add a similarity-pruning baseline, Slim-SC, in the camera-ready version.
>
> **W3: Wall-clock accounting under large batches/multi-GPU, and periodic probing vs. achieved savings**
>
> We appreciate the reviewer’s comment. We believe that DUCHESS's gain will be consistent across different inference systems and hardware settings due to its inherent nature: shorter branches and parallel branch execution. We will provide measurements across different batch sizes and on multi-GPU testbeds in the camera-ready version.
>
> **Q1: Sensitivity to multiple DUCHESS hyperparameters and possible autotuning procedures.**
>
> We added sensitivity analysis of DUCHESS to $\tau, i, S$ in section 2.2 of the general response. We provided clarification of hyperparameter optimization strategies and discussed potential autotuning in section 3.2 of the general response.
>
> **Q2: Does a probe trained on one backbone/size transfer to others?**
>
> We believe that per-model training is required. Different LLM backbones have varying hidden dimension sizes and may have distinct activation distributions. Overall, it is difficult to directly apply the MLP probe trained for one backbone onto another without any tuning or adaptations.
>
> **Q3: Ablate early termination vs. branch-level pruning: what fraction of latency/tokens each contributes?**
>
> We added experimental results for ablating early termination and branch orchestrations in section 2.1 of the general response. At 10 branches, early termination alone reduces tokens by 30% compared to default self-consistency, and adding branch orchestration reduces them by an additional 12.3%, for the GSM8K/R1-Distill-Llama use case. We are running latency measurements and will update the rebuttal once we have results.
>
> **W4, Q4: Starvation of hard requests under difficulty-aware scheduling**
>
> We thank the reviewer for raising this important point. A straightforward mechanism is to incorporate a priority score that combines predicted request difficulty with a starvation term that increases with wait time. This prevents hard requests from being indefinitely deprioritized under load. We plan to include this mechanism and its analysis in the camera-ready version.

---

### Official Review · Reviewer_kRNj · 2025-11-02

**Soundness:** 3
**Presentation:** 2
**Contribution:** 3
**Rating:** 6
**Confidence:** 2

**Summary:**

In this paper, the authors study the problem of LLM serving system. In particular, they propose an LLM serving system which they call DUCHESS that orchestrates multi-branch reasoning within a single request using predictions from a lightweight probe over LLM activations. DUCHESS takes one of three branch-level actions:

- (1) Early termination with CoT probing to elicit a short final answer
- (2) Selective branch-out (forking promising branches while reusing KV cache)
- (3) Continuation (keep generating tokens for branches that are still uncertain, giving them additional steps to potentially reach a correct answer)

Beyond intra-request control, the system optionally adds complexity-aware scheduling across a request pool, which using prefill activations of the first decoded token to predict difficulty, the scheduler serves easier prompts first to reduce mean latency.

Experiments on GSM8K, MMLU, and MATH with DeepSeek-R1-Distill-Llama-8B and vLLM, show effectiveness under both cost–accuracy trade-offs (tokens vs. accuracy) and serving latency.

Admittedly, the area of this research is not my primary research area. I apologise that I can only share some general comments on this paper.

**Strengths:**

``S1``: The empirical gains on both cost–accuracy and serving latency look good, with clear and reproducible setups. Particularly,

- ``reducing token usages by 42%-63% at matched
accuracy compared to Self-consistency``,
- ``reducing mean, P50, and
P95 latencies by 57-81%, 58-85%, and 52-84% on three datasets``, and
- ``reducing mean latency by up to 29.7% and 34.7%
compared to FCFS``.

``S2``: Selective branch-out with KV reuse is a practical systems optimisation that preserves parallelism and reduces stragglers.

**Weaknesses:**

I apologise that I’m not very familiar with LLM serving systems. I will carefully refer to the comments of other fellow reviewers, who are more confident than me, for my final judgment. Some general weaknesses that I can notice are as follows.

``W1``: The authors currently focus on math plus general knowledge (GSM8K, MMLU, MATH). A broader commonsense and graduate-level domain generalization is in fact, not tested. The inclusion of SuperGPQA and CommonsenseQA would help strengthen the contributions of the proposed LLM serving system DUCHESS.

``W2``: Request termination and voting don't actually account for dependencies introduced by branch duplication. I have noticed that the authors mention this as a limitation at the end of the paper. This opens the door to weighted voting or de-duplication.

**Questions:**

Apart from those aforementioned in the weaknesses section:

``Q1``: I wonder would selective branch-out reduce reasoning diversity?

``Q2``: The authors report 16.4% and 7.8% mistaken early termination on MMLU/MATH. It would be great if the authors could elaborate the possibility of a detecting and recovering mechanism based on this finding.

---

> ### Author Response · Authors · 2025-11-22
> **Author's Response**
>
> We thank the reviewer for the comments, especially acknowledging our system-friendly designs, such as KV reuse in branch duplication.
>
> **W1: Including broader evaluation datasets.**
>
> We have added the GPQA dataset. Please refer to section 1.2 of the general response for the results. At 10 branches per request, DUCHESS achieves the highest accuracy while using the fewest tokens compared to default self-consistency and other baseline methods. The mean latency and TTFTs are reduced by 69% and 86%, respectively, relative to default self-consistency.
>
> We also evaluated additional LLMs and baseline methods to demonstrate the generalizability of DUCHESS. Please check section 1 of the general response for details.
>
> **W2: Request termination and voting don’t account for dependencies introduced by branch duplication.**
>
> A natural solution to solve the dependencies created by branch duplication is to introduce a **weighted voting mechanism**. Each branch starts with an initial weight, and at every branch-out this weight is redistributed among the resulting branches. When a branch terminates early, we record its weight and incorporate it into the final vote, reducing bias. As the reviewer noted, we briefly mentioned this in the limitations section.
>
> As future work, we plan to investigate: (1) how branch weights should decay after each branch-out, and (2) how to combine these weights with predicted correctness probabilities for early-terminated answers.
>
> **Q1: Selective branch-out reduces reasoning diversity.**
>
> The selective branch-out process is triggered only when a branch is early or naturally terminated. Early termination happens when branches exhibit clearly different qualities (for example, some are on a correct problem-solving trajectory while others have already diverged or made an error). When **no** branches are sufficiently “promising,” early termination simply does not occur, and all branches continue forward, which preserves reasoning diversity. In practice, we set a high early-termination threshold (e.g., $\geq$ 70th percentile on validation-set predictions) and require **S consecutive predictions above this threshold** before ending a branch early. Thus, selective branch-out only happens when a branch is extremely promising with consistently high confidence. As a result, the risk of reducing reasoning diversity is low.
>
> **Q2: Detecting and recovering mistaken early-terminated branches.**
>
> We thank the reviewer for this insightful question. Indeed, mistaken early termination can be mitigated by adding a detection-and-recovery step. Specifically, for any early-terminated branch, we can ask the LLM to generate **multiple answers**; if these answers disagree substantially, we simply **resume** that branch instead of terminating it. This multi-shot mechanism can reduce erroneous early terminations and improve overall accuracy. (Our current system uses a one-shot answer for early-terminated branches.) We will mention this direction in the camera-ready version and plan to implement it in future work.

---

### Author Response · Authors · 2025-11-22
**General Response (Part 1/7)**

# General response

We thank the reviewers for their thoughtful feedback. We appreciate that the reviewers unanimously acknowledge the practical relevance of DUCHESS, including substantial cost and latency reductions at matched accuracy for multi-branch reasoning inference (all reviewers), and a lightweight, efficient system design (kRNj: “Branch-out with KV reuse is a practical systems optimisation that preserves parallelism and reduces stragglers, ”, cMbD: “a hardware/system aware design”, XiJK: “lightweight prediction and minimal overhead”), with well-grounded and intuitive motivations (cMbD: “helpful layer-wise probe analysis”, GqzD: “core intuition and methodology are well-grounded in observations from prior research work”). Our evaluation of serving latencies and TTFTs under vLLM with varying loads further demonstrates practicability in real production environments (cMbD, GqzD). Reviewers thought the paper was written succinctly with informative diagrams (GqzD). Overall, we appreciate the recognition of our contributions to efficient LLM serving.

However, reviewers raised concerns about whether our results generalize across additional LLMs and datasets, and about the lack of baseline methods. In particular, reviewers desired (1) evaluations using graduate-level benchmarks (kRNj) that typically involve longer reasoning traces (GqzD), (2) experiments with other reasoning and non-reasoning LLMs (GqzD), and (3) comparisons to existing incremental or entropy-aware self-consistency methods, such as ESC, which reduces reasoning token usage but not latency (cMbD, GqzD). Further, reviewers had questions about the hyperparameter tuning process of the MLP (GqzD) used in per-branch correctness prediction as well as the intra-branch orchestration policy (cMbD, GqzD). Reviewers desired a more detailed analysis of our proposed approach, including ablating system components (cMbD) and sensitivity to various hyperparameters (cMbD, GqzD).

Our additional experiments demonstrate that DUCHESS's performance excels across multiple LLMs and datasets, over more existing methods. The rest of the general response gives details regarding the experiments, and covers other issues such as ablations and sensitivity analysis. We also provide individual responses to each reviewer.
- Section 1: additional evaluations with more LLMs, dataset and baseline method. Overall, DUCHESS consistently achieves gains in cost and latency, demonstration generalizability across diverse use cases.
    - Section 1.1: additional reasoning LLM (Intern-S1) and non-reasoning LLM (Gemma-3)
    - Section 1.2: additional dataset GPQA
    - Section 1.3: additional baseline incremental self-consistency method, early self-consistency (ESC)
- Section 2: additional analysis for gain calibration
    - Section 2.1: ablation of early termination and branch orchestration.
    - Section 2.2: sensitivity analysis to hyperparameter choices
    - Section 2.3 sensitivity to selected LLM layer
- Section 3: clarifications
    - Section 3.1: MLP architecture settings for per-branch correctness prediction task
    - Section 3.2: DUCHESS hyperparameter optimizations and possible autotuning procedures

We will incorporate these changes to the paper and submit a revised PDF during the rebuttal period. We believe that we have been able to address all concerns with our additional experiments (with a few in-progress ones that we plan to provide results for as soon as possible), along with our per-review replies.

---

> ### Author Response · Authors · 2025-11-22
> **General Response (Part 2/7)**
>
> ## 1 Additional Evaluations
>
> ### 1.1 Additional LLMs
>
> We added two additional LLMs: (1) [internlm/Intern-S1-Mini](https://huggingface.co/internlm/Intern-S1-mini) , an 8B reasoning model with 36 transformer layers, and (2) [google/gemma-3-4b-it](https://huggingface.co/google/gemma-3-4b-it), an instruction-tuned, general-purpose model with 34 layers. These results demonstrate that our proposed technique generalizes to broader classes of reasoning and non-reasoning LLMs.
>
> #### 1.1.1 Layer-wise probing
>
> Following the settings in the original submission, we report the validation and test accuracy of per-branch correctness predictions on GSM8K, trained on activations from different LLM layers. For both models, middle layers yield the highest accuracies, consistent with our findings from the R1-Distill-Llama model.
>
> [Figure: Per-layer probing for Intern-S1](https://i.ibb.co/KpsY7pnX/pred-acc-by-layer-internlm.png)
>
> [Figure: Per-layer probing for Gemma-3](https://i.ibb.co/vnPGXKw/pred-acc-by-layer-gemma.jpg)
>
> We then use the layers with the highest validation accuracy (21 and 17 for Intern-S1 and Gemma, respectively) for the other benchmarks: MMLU and MATH.
>
> #### 1.1.2 Cost-accuracy tradeoffs
>
> Due to time and resource limitations in the rebuttal phase, we report the computational cost (average tokens generated per request) and accuracy at 10 branches per request. We plan to include full pareto-frontier comparisons in the camera-ready version. The numbers are averaged from 10 trials, with variances omitted for space considerations. DUCHESS consistently reduces tokens while matching or surpassing the accuracy of default self-consistency in all settings, demonstrating generalizability across model classes.
>
> We **bold** the lowest cost numbers that matches Default SC accuracies, and **bold** highest accuracies. For Intern-S1:
> | method          |   GSM8K tokens |   GSMK accuracy |   MMLU tokens |   MMLU accuracy |   MATH tokens |   MATH accuracy |
> |:---------------|---------------:|-----------------:|--------------:|----------------:|--------------:|----------------:|
> | Default SC     |        3059.87 |            0.925 |      10026.4  |           0.816 |       7309.04 |           0.702 |
> | Dynasor        |        3436.24 |            **0.928** |       6746.96 |           0.815 |       6960.69 |           0.674 |
> | Short-m@k      |        2866.53 |            0.916 |       6986.62 |           0.766 |       6430.16 |           0.662 |
> | DUCHESS (Ours) |        **2508.9**  |            0.925 |       **4620.95** |           **0.821** |       **6044.24** |           **0.709** |
>
> For Gemma-3:
> | method          |   GSM8K tokens |   GSM8K accuracy |   MMLU tokens |   MMLU accuracy |   MATH tokens |   MATH accuracy |
> |:---------------|---------------:|-----------------:|--------------:|----------------:|--------------:|----------------:|
> | Default SC     |        4006.29 |            **0.91** |       3539.38 |           0.658 |       9127.72 |           0.701 |
> | Dynasor        |        3525.18 |            0.9 |       3116.49 |           0.64  |       7567.99 |           0.663 |
> | Short-m@k      |        3600.76 |            **0.91** |       3356.97 |           0.66  |       8330.42 |           0.701 |
> | DUCHESS (Ours) |        **3136.78** |            **0.91**  |       **2863.73** |           **0.667** |       **7829.84** |           **0.704** |
>
> Both Intern-S1 and Gemma-3 generate significantly shorter responses than R1-Distill-Llama (e.g., at 10 branches, Gemma-3 takes 3136.8 tokens per request on GSM8K, while R1-Distill-Llama takes 7763.4 tokens per request, as shown in Figure 3 in the submission). This may be due to R1-Distill-Llama naturally outputs its internal “thinking” processes, while Intern-S1 and Gemma-3 require COT-styled prompting. Shorter responses reduce room for token pruning; nevertheless, DUCHESS still achieves sizable cost reductions.
>
> We observe that the Dynasor baseline method increases token usage compared to default self-consistency for the Intern-S1 and GSM8K setting. Dynasor frequently queries the LLM for intermediate answers, which costs 10 additional tokens per 64 decoding steps. For shorter responses, frequent answer extractions inflate token usage and outweigh gains from early terminations.

---

> ### Author Response · Authors · 2025-11-22
> **General Response (Part 3/7)**
>
> #### 1.1.3 Serving latencies and TTFTs
>
> We first evaluate serving latencies and TTFTs under normal request arrival rates. Then we report latency as a function of increasing rates to demonstrate robustness. The vLLM version is upgraded to 0.11.0 for these experiments to comply with LLM requirements.
>
> Matching the original submission, we randomly generate request arrival schedules following Poisson distributions. For Intern-S1, we set the average queries-per-minute (QPM) to be 10/2/4 on GSM8K/MMLU/MATH respectively, based on the upper quantile of Default SC processing time per request: 6.2/32.1/16.8 seconds. For Gemma-3, the average QPM is 10/15/3 based on the upper quantiles of Default SC processing time per request: 6.7/4.7/19.6 seconds, on GSM8K/MMLU/MATH, respectively.
>
> Latency numbers for Intern-S1 (**bolded** numbers are the best):
> | method |   GSM8K Mean |   GSM8K P50 | GSM8K P95 |   MMLU Mean |   MMLU P50 |   MMLU P95 |   MATH Mean |   MATH P50 |   MATH P95 |
> |:--------|-------------:|------------:|------------:|------------:|-----------:|-----------:|------------:|-----------:|-----------:|
> | Default SC |         38.2 |        31.9 |        85.1 |        75.4 |       58.7 |      198.8 |       263.2 |      267.6 |      459.1 |
> | Short-m@k  |         11.5 |         8.7 |        31.6 |        15.4 |       10.1 |       **46.5** |        37.5 |       17.7 |      130.3 |
> | DUCHESS (Ours) |          **9.3** |         **7.2** |        **24.2** |        **14.9** |        **7.8** |       58.8 |        **29.5** |       **13.2** |       **98.1** |
>
> TTFT numbers for Intern-S1:
> | method |   GSM8K Mean |   GSM8K P50 | GSM8K P95 |   MMLU Mean |   MMLU P50 |   MMLU P95 |   MATH Mean |   MATH P50 |   MATH P95 |
> |:--------|-------------:|------------:|------------:|------------:|-----------:|-----------:|------------:|-----------:|-----------:|
> | Default SC |        32.79 |       26.64 |       79.21 |       53.14 |      35.07 |     170.42 |      246.28 |     248.21 |     433.34 |
> | Short-m@k  |         7.23 |        4.2  |       27.44 |        **4.92** |       **0.05** |      **27.02** |       26.67 |       6.42 |     117.08 |
> | DUCHESS (Ours) |         **5.32** |        **3.06** |       **19.8**  |        5.91 |       **0.05** |      39.05 |       **20.06** |       **3.03** |      **87.6**  |
>
> Short-m@k achieves lowest tail latencies and all TTFTs on MMLU, however, it reduces accuracy from Default SC by a large margin (-0.05). In contrast, DUCHESS slightly improves the accuracy (+0.005) with lower latencies.
>
> Latency numbers for Gemma-3:
> | method |   GSM8K Mean |   GSM8K P50 | GSM8K P95 |   MMLU Mean |   MMLU P50 |   MMLU P95 |   MATH Mean |   MATH P50 |   MATH P95 |
> |:--------|-------------:|------------:|------------:|------------:|-----------:|-----------:|------------:|-----------:|-----------:|
> | Default SC |       136.16 |      139.77 |      247.93 |      302.72 |     283.07 |     495.69 |       36.48 |      28.12 |     100.23 |
> | Short-m@k  |        12.15 |        8.63 |       37.06 |       24.77 |      23.02 |      71.38 |       17.87 |      14.17 |      48.81 |
> | DUCHESS (Ours) |         **8.84** |        **5.46** |       **28.91** |       **14.16** |      **11.65** |      **35.52** |       **14.35** |      **11.25** |      **37.41** |
>
> TTFT numbers for Gemma-3:
> | method |   GSM8K Mean |   GSM8K P50 | GSM8K P95 |   MMLU Mean |   MMLU P50 |   MMLU P95 |   MATH Mean |   MATH P50 |   MATH P95 |
> |:--------|-------------:|------------:|------------:|------------:|-----------:|-----------:|------------:|-----------:|-----------:|
> | Default SC |       130.13 |      132.03 |      241.75 |      298.24 |     278.62 |     491.96 |       22.79 |      12.7  |      82.27 |
> | Short-m@k  |         8.05 |        4.02 |       29.93 |       21.06 |      18.83 |      67.45 |        7.84 |       0.11 |      36.11 |
> | DUCHESS (Ours) |         **5.26** |        **1.57** |       **23.97** |       **10.84** |       **8.05** |      **32.7**  |        **5.51** |       **0.04** |      **24.59** |
>
> We then report the mean/P50/P90 latencies as a functions of request arrival rates, for Intern-S1:
>
> [Figure: Latency as a funcrtion of QPM for Intern-S1](https://i.ibb.co/CZW5S6b/latency-qpm-intern.png)
>
> And Gemma-3:
>
> [Figure: Latency as a funcrtion of QPM for Gemma-3](https://i.ibb.co/wF1n9t1H/latency-qpm-gemma.png)
>
> We conclude that DUCHESS consistently reduces reasoning cost and latency across broad model classes.

---

> ### Author Response · Authors · 2025-11-22
> **General Response (Part 4/7)**
>
> ### 1.2 Additional Dataset
>
> We include the [GPQA](https://huggingface.co/datasets/Idavidrein/gpqa) dataset as requested by the reviewers (kRNj, GqzD). Due to time and resource constraints, we report the cost/accuracy trade-off and latency numbers only for R1-Distill-Llama at 10 branches per request. For the camera-ready version, we will complete this evaluation with (1) a full Pareto-frontier comparison for cost-accuracy tradeoffs, and (2) latencies at varying request arrival rates.
>
> Based on results from the submission (Figure 2), we select layer 14’s activation for correctness predictions. However, since this training set is highly imbalanced (most questions are answered incorrectly), we use the validation F1 score instead of accuracy as the metric when optimizing MLP hyperparameters. Each branch is capped at 4,096 tokens. Results are averaged from 10 trials with errors omitted for space.
>
> #### 1.2.1 Cost-accuracy tradeoffs
>
> |                |   GPQA cost (tokens/request) |   GPQA accuracy |
> |:---------------|--------------:|----------------:|
> | Default SC     |       6970.17 |            0.28 |
> | Dynasor        |       6321.88 |            0.29 |
> | Short-m@k      |       4862.62 |            0.3  |
> | DUCHESS (Ours) |       **4481.99** |            **0.31** |
>
> Default SC archives low accuracy on the GPQA dataset. [DeepSeek's official huggingface page](https://huggingface.co/deepseek-ai/DeepSeek-R1) reports 0.49 accuracy on the Diamond subset, using 64 branches and a maximum of 32,768 tokens per branch. Our accuracy gap most likely comes from fewer branches with shorter lengths. For the camera-ready version, we would like to extend our branch and max token settings to match the official version. Overthinking contributes to a significant portion of the mistakes: among all branches with incorrect final answers, 40.7% had at least one correct intermediate answer but switched in the end.
>
> DUCHESS achieves the highest accuracy, with 0.03 points (approximately 11%) higher than the default SC. This demonstrates DUCHESS can successfully prevent LLM overthinking by combining (1) branch early terminations and (1) expanding “promising” branches with higher predictions. DUCHESS has the lowest token cost among the techniques, with approximately a 36% reduction from the default SC.
>
> #### 1.2.2 Latencies and TTFTs
>
> We randomly generate a request arrival schedule following a poisson distribution with an average of 3 queries per minute (decided by the upper quantile of default SC’s average processing time per request: 18 seconds). We plan to include measurements at different QPMs in the camera-ready version. We include short-m for reference, although the accuracy is unmatched.
>
> Latency numbers:
> |         |   GPQA Mean |   GPQA P50 |   GPQA P95 |
> |:--------|------------:|-----------:|-----------:|
> | Default SC |       36.39 |      21.95 |     124.41 |
> | Short-m@k  |       12.39 |       8.66 |      33.82 |
> | DUCHESS (Ours) |       **10.89** |       **7.57** |      **30.82** |
>
> TTFT numbers:
> |         |   GPQA Mean |   GPQA P50 |   GPQA P95 |
> |:--------|------------:|-----------:|-----------:|
> | Default SC |       20.92 |       5.25 |     106.96 |
> | Short-m@k  |        3.36 |       **0.04** |      23.68 |
> | DUCHESS (Ours) |        **3.31** |       **0.04** |      **18.74** |
>
> DUCHESS archives 70%, 65%, 75% reduction in the mean, median, and tail latencies, along with 84%, 99%, 82% reductions in the mean, median, and tail TTFTs. The gains are similar to the datasets evaluated in the original submission. Our results demonstrate that DUCHESS is able to generalize across multiple tasks.

---

> ### Author Response · Authors · 2025-11-22
> **General Response (Part 5/7)**
>
> ### 1.3 Additional Baseline Method
>
> We include [ESC: Early-Stopping Self-Consistency](https://openreview.net/pdf?id=ndR8Ytrzhh), published at ICLR'24, as an additional baseline method. ESC reduces the number of branches per request by generating w branches at a time; then, if answers from the last w branches agree (i.e., zero entropy), ESC terminates the request. Otherwise, it generates w more branches until exhausting a pre-defined maximum branches per request. We add ESC for R1-Distill-Llama & GSM8K evaluation only, and plan to extend this evaluation to more models/datasets in the camera-ready version.
>
> We set w as 2 and the maximum branches per request as 10 for ESC. We fixed DUCHESS at 10 branches per request. The token/accuracy results are averaged from 10 trials with errors omitted for space.
>
> |                |   GSM8K cost (tokens/request) |   GSM8K accuracy |
> |:---------------|---------------:|-----------------:|
> | Default SC     |        7763.37 |             0.8  |
> | ESC            |        **2860.58** |       **0.83** |
> | DUCHESS (Ours) |        4473.53 |             0.8  |
>
> ESC is more optimal for cost-accuracy tradeoffs than DUCHESS with fewer tokens and higher accuracy. We then report latencies and TTFTs with an average load of 2 queries per minute (matching the original submission).
>
> Latencies:
> |         |   GSM8K Mean |   GSM8K P50 |   GSM8K P95 |
> |:--------|-------------:|------------:|------------:|
> | Default SC |        54.05 |       40.82 |      156.09 |
> | ESC     |       147.13 |       85.07 |      454.95 |
> | DUCHESS (Ours) |        **10.48** |        **7.68** |       **24.69** |
>
> TTFTs:
> |         |   GSM8K Mean |   GSM8K P50 |   GSM8K P95 |
> |:--------|-------------:|------------:|------------:|
> | Default SC |        34.2  |       16.13 |      134.53 |
> | ESC     |       123.5  |       59.95 |      432.68 |
> | DUCHESS (Ours) |         **2.76** |        **0.07** |       **14.43** |
>
> ESC increases mean latency by 2.7X and 14.1X under normal request loads, compared to Default SC and DUCHESS, respectively. This is because ESC generates branches incrementally and increases per-request completion time. In contrast, DUCHESS’s latency-aware design ensures fully parallelizable branch execution, ultimately achieving consistent savings in both cost and latency metrics compared to default SC.

---

> ### Author Response · Authors · 2025-11-22
> **General Response (Part 6/7)**
>
> ## 2 Analysis
>
> Due to time constraints in the rebuttal period, our additional analysis is only on the R1-Distill-Llama model and GSM8K dataset. We plan to extend these analyses to more models and datasets in the camera-ready version.
>
> ### 2.1 Ablation of early termination and branch orchestration
>
> We add a variant of DUCHESS that terminates branches early based on MLP predictions, a threshold, and S (the number of consecutive predictions >= threshold required to terminate a branch early), but without any cross-branch optimizations (e.g., selective branch-out). The remaining settings are consistent with Figure 2 in the submission: we vary the number of branches per request between 5 and 10 inclusively. Markers within a curve correspond to increasing number of branches per request from left to right. The results are averaged from 10 trials, with error bars representing 95% confidence intervals.
>
> [Figure: Early termination and branch orchestration ablation.](https://i.ibb.co/R47RnrPL/token-accuracy-ablation.png)
>
> We see that at fewer branches per request, only performing early termination achieves better Pareto-optimality than including branch orchestration. However, as the number of branches increases, expanding on the more “promising” branches can lead to a faster majority with high accuracy: at 10 branches per request (the right-most marker on each curve), early termination alone reduces tokens by 30% compared to default self-consistency, and adding branch orchestration reduces them by an additional 12.3%. We are working on provide latency measurements and will update the rebuttal once we have results.
>
> ### 2.2 Sensitivity to DUCHESS hyperparameters
>
> We analyze DUCHESS sensitivity to three key hyperparameters: $\tau$ (threshold for early termination), $i$ (probing interval), and $\S$ (number of probes with prediction $\geq \tau$ required for early termination).
>
> [Figure: Sensitivity analysis of DUCHESS hyperparameters](https://i.ibb.co/JRb3sY9q/sensitivity-analysis.jpg)
>
> The figure above plots the cost (shown in blue on the left y-axis) and accuracy (shown in red on the right y-axis) as functions of different hyperparameter values. Results are reported on GSM8K/R1-Distill-Llama at 10 branches per request. For each subplot, we vary one hyperparameter (marked on the x-axis) while keeping the others at their selected values: 70/16/2 for $\tau$/$i$/$S$, which gives optimal validation cost-accuracy trade-offs.
>
> For all three hyperparameters, smaller values reduce token cost per request. This aligns with expectation, since smaller $\tau$ and $S$ values lower the early-termination requirements, whereas smaller $i$ values increase probing frequency. Overly small $\tau$ and $S$ values reduce accuracy by up to $0.024$. However, excessively large values only inflate token usage without accuracy gains. Accuracy varies non-monotonically with respect to $i$, as overly frequent probing may aggressively terminate branches and cause accuracy drops, while probing too rarely reduces the chances of preventing CoT overthinkings.
>
> ### 2.3 Sensitivity to selected LLM layer
>
> In the original submission, we selected layer 14 based on validation performance on a single dataset and model setting. Ideally, we would profile layer-wise accuracy for each dataset and model use cases, but constrained by time and resources.
>
> Our key observation is that middle layers are more informative, hence better for correctness prediction than other layers. This observation is validated on two additional LLMs (see rebuttal section 1.1.1). Our method is robust to the specific layer choice, as long as a sufficiently accurate layer (i.e., one in the middle range) is selected. To demonstrate the robustness, we show the cost-accuracy tradeoff of R1-Distill-Llama on GSM8K, using predictions from layers 11 (middle-left), 14 (optimal), and 22 (middle-right) of the transformer.
>
> [Figure: Sensitivity to layer selection](https://i.ibb.co/zVkdcBnH/sensitivity-to-layers.png)
>
> Overall, L11 and L22 perform closely to L14 in terms of token usage. The accuracy gap is at most 0.01, which could be reduced by optimizing hyperparameters specific to each layer choice.

---

> ### Author Response · Authors · 2025-11-22
> **General Response (Part 7/7)**
>
> ## 3 Clarifications
>
> ### 3.1 MLP architecture setting
> In lines 192-193 of the original submission, we stated that our predictor design has “minor dataset-specific adjustments based on validation performance”. We clarify our MLP designs here:
>
> **Selection of LLM layers.** We ideally would profile layer-wise accuracy for each dataset and LLM use cases, but are constrained by time and resources. We thus profile the layer-wise accuracy of each model on the GSM8K dataset and identify the optimal layer that maximizes validation accuracy (layer 14, 21, and 17 for R1-Distill-LLama, Intern-S1, and Gemma-3, respectively) to use on the remaining benchmarks.
>
> **MLP general architecture.** The MLP's input layer has a dimension equivalent to the LLM hidden size (4,096 for R1-Distill-Llama and Intern-S1, and 2,560 for Gemma-3), followed by a LayerNorm, then a series of hidden fully-connected (FC) layers, and finally an output layer of dimension 1, corresponding to the logit of probability of correctness.
>
> **Training configurations.** The MLPs are optimized using AdamW with an initial learning rate of $3\times10^{-4}$ and 0.1 weight decay. We adjust the learning rate over epochs using StepLR scheduler with step size 5 and gamma 0.8. We use the Binary Cross-Entropy loss function, appropriate for classification tasks. We train the model for up to 15 epochs, and keep the model weights from the epoch with the highest validation accuracy.
>
> **Per-dataset adjustments.** We vary a few MLP architectural and training configurations per dataset and LLM to maximize their validation accuracies. The MLP architectural settings include: the number of FC layers (2-5), the dimension of each FC layer (selected from $[256, 512, 1024, 2048, 4096]$), and the dropout rate on FC layers (0.1-0.3). The training configuration changes include positive class weights (inversely proportional to the fraction of positive labels in training data) and label smoothing (0-0.1). Adjusting these settings specific to each dataset is necessary, as the datasets vary in cardinality and label distributions.
>
> We predefine a search range per MLP setting: e.g., the number of hidden layers is set to 2 to 5, and the hidden layer size is chosen from [256, 512, 1024, 2048, 4096]. Then we perform a grid search, with maximizing validation accuracy as the objective. To reduce the search space, we change one setting at a time while keeping the others fixed, and restrict hidden-layer sizes to monotonically decreasing values.
>
> ### 3.2 DUCHESS hyperparameters
> In lines 726-727 of Appendix A.1 of the submission, we stated that DUCHESS hyperparameters are optimized based on cost-accuracy trade-offs on the validation set. That is, to minimize token usage costs while achieving accuracy that matches or exceeds Default SC. We then use the same settings for latency evaluations. We do not tune hyperparameters based on latency results, as DUCHESS's latency-aware design (i.e., parallel executions, KV reuse from forked branches, straggler preemption) ensures that low token cost leads to low request serving latencies.
>
> Let $c$ and $P$ represent the number of branches per request and a hyperparameter configuration for DUCHESS's. Let $Cost(c, P)$ and $Acc(c, P)$ be the cost (token usage) and accuracy of DUCHESS's given specific $c$ and $P$ values on the validation set. Let $AccSC(c)$ be the validation accuracy of default self-consistency at $c$ branches per request. Then, the maximum accuracy achieved by Default SC over all possible $c$ values is:
>
> $A^\*=max_{c\in C} AccSC(c)$
>
> Then, the cost-accuracy tradeoff for hyperparameter values $P$ is given by:
>
> $OBJ(P, A^\*) = \max_{c \in C} (M - Cost(c, P))\cdot I(Acc(c, P) \geq A^\*)$
>
> Where $M$ is the upper bound of cost per request, and $I(Acc(c, P) \geq A^\*)$ is an indicator variable with a value of $1$ if $Acc(c, P) \geq A^\*$, and a value of $0$ otherwise. Finally, the optimal parameter choice, $P^\*$ is given by:
>
> $P^\* = \text{argmax}_P OBJ(P, A^\*)$
>
> **Auto-tuning.** Because of the well-formulated objective function, DUCHESS's hyperparameter tuning can be effectively automated by optimization algorithms such as grid search and Bayesian optimization~\citep{snoek2012practical}. In this work, we used a one-parameter-at-a-time grid search to reduce the search space.

---

### Author Response · Authors · 2025-11-27
**Summary of rebuttal**

We thank the reviewers for their thoughtful feedback, which helps us strengthen our work in this rebuttal. Reviewers consistently recognized the practical relevance of DUCHESS, highlighting its substantial cost and latency reductions at matched accuracy for multi-branch reasoning inference. They further emphasized our lightweight, hardware-/system-aware design: kRNj noted that “branch-out with KV reuse is a practical systems optimization that preserves parallelism and reduces stragglers,” cMbD highlighted the “hardware/system-aware design.” Xijk noted the “lightweight prediction with minimal overhead.” Reviewers also found our motivations and analysis well-grounded, such as the helpful layer-wise probe study (cMbD) and strong methodological intuition grounded in prior work (GqzD). Our evaluation of serving latencies and TTFTs under vLLM with varying loads demonstrates real-world practicality (cMbD, GqzD). The reviewer additionally appreciated the paper’s succinct writing and informative diagrams (GqzD). We are grateful for the recognition of our contributions to efficient LLM serving.

However, reviewers desired more evaluation results using other reasoning and non-reasoning LLMs, datasets, and comparisons to cost-aware incremental self-consistency methods (kRNj, cMBd, GqzD). Reviewers were concerned about the generalizability of our complexity-aware request scheduler (GqzD, XiJK). Reviewers also asked for clarifications of multiple system designs and hyperparameter choices (cMBd, GqzD). Finally, Reviewers desired a more detailed analysis of our proposed approach, including ablating system components (cMbD) and sensitivity to various hyperparameters (cMbD, GqzD).

We have improved our paper in response to the reviews during the rebuttal. We believe that our revised paper, along with per-reviewer replies, resolves most concerns. Our changes are marked in purple in the revised PDF:

- Section 3.1 Per-branch correctness prediction

We clarified our predictor architectural designs, training configurations, and per-dataset variations (cMBd, GqzD).
- Section 3.2 Intra-request branch orchestration

We clarified the definitions, choices, and tuning procedures for hyperparameters (cMBd, GqzD).
- Section 3.3 Inter-request complexity-aware scheduling

We extended the complexity-aware scheduling to datasets without pre-labeled request complexities. Our request scheduler now generalizes to all datasets (GqzD, XiJK).
- Section 4.1 Experimental settings

We clarified our experimental settings and added descriptions to the new LLMs and datasets added in the rebuttal period (kRNj, cMBd, GqzD).
- Section 4.2 Evaluation results

We added new evaluation results on the cost-accuracy tradeoffs and serving latencies/TTFTs, using additional LLMs (GqzD) and datasets (kRNj, GqzD). Our results demonstrate that our proposed technique generalizes to broad classes of reasoning and non-reasoning LLMs and across multiple datasets.
- Section 4.3 Inter-request complexity-aware scheduling

We evaluated the inter-request complexity-aware scheduling on GSM8K and MMLU datasets, using LLM-labeled request complexity as groundtruths to train the request complexity predictor. We show that the scheduler remains effective for datasets without provided request complexities (GqzD, XiJK).
- Section 4.4 Comparison to cost-aware self-consistency

We compare DUCHESS to an incremental self-consistency baseline: Early-stopping Self-Consistency (ESC) as requested by multiple reviewers (cMBd, GqzD). We show that ESC reduces token costs but inflates serving latencies. In contrast, DUCHESS reduces both cost and latency while maintaining accuracy.
- Section 4.5 Analysis

We added discussions on error recovery from mistaken early terminations, using a multi-shot mechanism (kRNj). We added new ablations of system components and sensitivity analysis of hyperparameters, and provided insights (cMBd, GqzD).

---

### Meta-Review · Area_Chair_WjoN · 2026-01-06

**Summary:**

**1) Summary**
This paper presents DUCHESS, a serving-time system for multi-branch chain-of-thought inference that uses a lightweight probe on intermediate activations to predict branch correctness and dynamically decide among early termination, selective branching, or continued generation. The system additionally includes an optional difficulty-aware scheduler to reduce mean latency under load. Experiments on GSM8K, MATH, and MMLU show large token savings and latency reductions at matched accuracy.

**2) Strengths**

* Introduces a unified serving-time orchestration policy (stop, fork, continue) grounded in lightweight correctness prediction over live activations.
* Provides strong empirical gains in both iso-accuracy token reduction and real-world serving latency, with system-level validation in vLLM.
* Demonstrates careful engineering integration—KV-cache reuse, mid-layer probing, periodic probing intervals—with low overhead.
* Includes optional inter-request complexity scheduling that further reduces tail latency and improves throughput.

**3) Weaknesses**

* Generalization is not thoroughly validated: probe training, layer choice, and hyperparameters are tuned on a single model/dataset, and cross-model robustness is unclear.
* Comparisons to prior multi-branch/early-stopping methods (e.g., ESC, RASC, Adaptive Consistency) are limited, making novelty and relative gains harder to assess.
* The connection between intra-request orchestration and inter-request scheduling is somewhat ad hoc, and scheduling fairness or starvation risks are under-analyzed.
* Applicability to tasks with longer reasoning traces or different model families is not demonstrated, leaving deployment generality uncertain.

**Reviewer Concerns:**

This is a very borderline paper and the acceptance/rejection decision is a hard one.

The reviewers did not follow up until the cutoff time, so it is really hard to determine if the concerns were fully addressed by the rebuttal. The authors made a substantial effort in addressing the concerns, and it seems that some of the concerns have been addressed, e.g., enhanced experimental evaluation on additional datasets, LLM models, etc.

On the other hand, the concerns of reviewers who held negative views seemed not fully addressed, e.g., the probe is trained per data/model, which makes generalization quite difficult. The authors promised to provide cross-dataset transferability analysis, but they did not seem to be able to provide it in the rebuttal.

Similarly, the hyperparameter selection was in a similar spirit and did not seem to be fully addressed.

Lastly, the confidence of the two negative reviews is higher than the two positive ones.

Given the above considerations, the AC tends to reject this paper.

**Reviewer Scores:**

The reviewers did not follow up to the cutoff time. So it is really hard to tell if the rebuttal would have changed the reviewers' mind. Based on the above consideration, it seems that they would not.

---

### Decision · Program_Chairs · 2026-01-26

Reject